# Surface Functionalization of (Pyrolytic) Carbon—An Overview

**Lucija Pustahija \* and Wolfgang Kern**

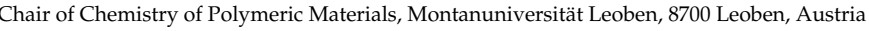

Chair of Chemistry of Polymeric Materials, Montanuniversität Leoben, 8700 Leoben, Austria
\* Correspondence: lucija.pustahija@unileoben.ac.at

**Abstract:** This review focuses on techniques for modifying the surface of carbon that is produced from sustainable resources, such as pyrolytic carbon. Many of these materials display high specific surface area and fine particle distribution. Functionalization of a surface is a commonly used approach in designing desired surface properties of the treated material while retaining its bulk properties. Usually, oxidation is a primary step in carbon functionalization. It can be performed as wet oxidation, which is a type of chemical surface modification. Wet oxidation is usually performed using nitric acid and hydrogen peroxide, as well as using hydrothermal and solvothermal oxidation. On the other side, dry oxidation is representative of physical surface modification. This method is based on corona discharge and plasma oxidation which are promising methods that are in line with green chemistry approaches. Whilst the oxidation of the carbon surface is a well-known method, other chemical modification techniques, including cycloadditions and various radical reactions on graphene layers, are presented as an alternative approach. Regarding secondary functionalization, coupling organosilanes to activated carbon is a common technique. Organosilanes bearing reactive groups present a bridge between inorganic species and polymer systems, e.g., epoxy and polyurethane resins, and facilitate the use of carbonaceous materials as reinforcing components for polymers and thermosetting resins. Along with the presented functionalization methods, this review also provides an overview of new applications of modified (i.e., functionalized) carbon materials, e.g., for the building industry, wastewater treatment, semiconducting materials and many more.

**Keywords:** carbon; pyrolytic carbon; surface functionalization; surface oxidation; coupling reactions; organosilanes; carbon composites

## 1. Introduction

Carbon materials have always been a matter of interest in designing novel materials and upgrading the properties of existing materials. It is well-documented how carbon improves the properties of materials. Regarding environmental demands, the sustainable production of carbon has become a topic of general interest.

One prominent way of producing carbon is via the pyrolysis of hydrocarbons (e.g., $CH_4$) which gives clean hydrogen gas as the major product [1,2]. Alternatively, carbon is produced by the pyrolysis of agricultural (waste) products. The major disadvantage of pyrolytic carbon is the potential contamination with catalysts. In addition, ways have to be found to make pyrolytic carbon compatible with other materials. Carbon black (CB) is commonly produced via pyrolysis. In its purest form, CB is a fine powder. Carbon black has a paracrystalline structure; this means its lattice is distorted with unit cells of highly variable shape and size [3]. Due to the latter, the functionalization of carbon black's surface presents a challenge in obtaining a uniformly modified surface. Once a proper approach has been found for surface modification, carbon can be properly dispersed in polymer matrices and thermosetting resins to give composites of high quality [4]. On the other hand, polymers are also used as binders for carbonaceous materials.

Usually, the oxidation of the surface of carbon constitutes the first step (primary functionalization,) in designing the desired surface before performing any secondary functionalization. Moreover, already well-known methods such as oxidation in various acids

and bases and the demand for safer, quicker and sustainable methods led to hydrothermal and solvothermal oxidation [5,6] and dry oxidation [5,6]. Dry oxidation with plasma and corona discharge are novel methods for primarily functionalizing powdery materials, including carbon.

The number and type of oxygen groups on the carbon surface are crucial for all further functionalization steps. The silanization of activated (i.e., oxidized) surfaces is an established method [7–10] and different silanes are used depending on the desired surface properties, including the presence of coupling units.

Furthermore, oxidation and subsequent condensation of organosilanes, and numerous additional methods are known to achieve a covalent functionalization of carbon surfaces. Prominent examples, which are rather found for the modification of graphene, carbon nanotubes and fullerenes (e.g., $C_{60}$) are based on cycloadditions of nitrenes, carbenes and ylides to the C=C bonds at the carbon surface.

In producing high-quality polymer composites, epoxy and polyurethane thermosets can be used with carbon [11–15]. Apart from traditional thermal curing of composites, progress has been achieved by curing with different sources of radiation (e.g., X-rays), and frontal polymerization using selected photo- and thermal initiators is a current topic of research [16–19].

The applications of functionalized carbons are not restricted to designing and producing novel polymer composites but are also found in the treatment of wastewater and polluted air, agriculture, storage of hydrogen, semiconductor materials and many more [20,21]. Furthermore, modified carbon materials can be used in the building industry [22–24]. In this paper, an overview of the properties of typical carbon materials is presented, with a focus on pyrolytic activated carbon. Thereafter, the most used primary and secondary functionalization methods are presented, together with the underlying reaction mechanisms. Finally, some characteristic applications of surface functionalized carbon are given.

Although many different methods of surface functionalization have already been explored for different types of carbon materials, we assume that they can also be used in the modification of pyrolytic carbon. This paper highlights and explains mechanisms of primary modification methods such as oxidation (with $HNO_3$, $H_2O_2$ and oxidation in plasma), secondary functionalization (with different organosilanes) and applications of functionalized carbon materials in epoxy and polyurethane composites. It is also worth mentioning that modification of carbon surface is not only limited to the oxidation and silanization methods. This general overview rather gives a short insight into what can be achieved with (different) carbon materials.

## 2. Properties and Characteristics of (Pyrolytic) Carbon

In the course of exploring the properties of novel materials, there is one material to which scientists are always coming back: carbon. In the last decades, carbon has been extensively studied as a novel material in various fields of applications because of its low weight, good mechanical properties and chemical resistance. To date, various (novel) carbon materials have been found, investigated and applied such as carbon nanotubes (CNTs), carbon fibres (CFs), carbon nanoribbons (CNRs), carbon onions (COs), carbon nanohorns (CNHs), etc. Because of rising emergencies caused by global warming, the production of hydrogen (as a fuel) is a current demand. One of the sustainable ways of producing hydrogen is the pyrolysis of hydrocarbons, which yields pyrolytic carbon as a valuable byproduct. In general, an inlet stream rich in organic species decomposes to give pyrolytic carbon and pyrolysis gases. Methane is mostly used as a raw material for the production of pure $H_2$ without $CO_2$ as a byproduct [25]. To achieve higher yields, various metal catalysts are used and—as a consequence—the produced carbon is usually contaminated with metal particles (e.g., nickel, cobalt, iron, aluminium, palladium, platinum, zirconium and chromium [26]). Different types of pyrolytic carbon can be produced, e.g., precursor materials for activated carbon [27]. One example of pyrolytic carbon is carbon black (CB)

which is rich in carbon (>97%) [28] and has a high surface area-to-volume ratio. CBs are usually produced by the pyrolysis of organic components or from agricultural waste materials, e.g., from coconut shells (pyrolysis of biomass) [29–32].

## 3. Functionalization of (Pyrolytic) Carbon

Because of their outstanding properties, carbon composite materials have been widely used in aerospace, medicine, sports equipment, the building industry, and many more. The major obstacle in using carbons is overcoming van der Waals intermolecular interactions that cause agglomeration and thus are responsible for the poor dispersibility in solvents and matrix resins [33]. A modification of the carbon surface can overcome these limitations.

The term "activation" of carbon refers to increasing the specific surface area by the formation of pores (via chemical and physical processes) [34]. The term "primary functionalization" refers to the enrichment of the carbon surface by the incorporation of oxygen groups, employing different oxidation strategies. As an example, the production of activated carbon from, e.g., biomass, usually occurs in a two-step process, which includes (i) volatilization of organic matter and (ii) activation of the carbon surface using thermal, physical or chemical methods. The processes are described in several references [27,35–39].

The term "surface modification" summarizes various ways of grafting different functional groups onto the carbon's surface by using physical or chemical processes [40–45]. According to Kato et al. [40], a "physical modification" can be performed by irradiation with electromagnetic waves and oxidation with gases. Physically functionalized surfaces usually suffer from low thermal stability and the inability to withstand a high load by shear forces. Moreover, functionalities introduced by physical functionalization methods can often be removed with chemicals [43]. On the other hand, a "chemical modification" can be performed with wet treatments, blending and metallization. Chemical modification is known as a convenient and controllable grafting method which leaves the bulk material unchanged [40].

Simply described, chemical functionalization can be divided into two groups, depending on which type of bonds are created: "covalent functionalization" and "noncovalent functionalization". Covalent functionalization is a destructive method by which $sp^2$-hybridized carbon atoms are attacked by reactive species and converted into $sp^3$-hybridized carbon atoms. It is the most developed method of grafting functional groups and it is usually preferred because of the ease of control. Chemical oxidation (i.e., wet oxidation) is usually the first step in the functionalization of an inert material's surface (i.e., primary functionalization).

Further functionalization of the carbon surface (i.e., secondary functionalization) is usually performed by grafting molecules onto the already oxidized surface. A characteristic example is the coupling of organosilanes which is dealt with in Section 3.2. Particular types of covalent functionalization are "grafting-to" and "grafting-from" reactions. The "grafting-to" reactions are based on the attachment of an end-functional group of a polymer to the surface. Various reactions can be used, e.g., amidation, esterification and radical coupling. Such reactions can be performed in a solution or a melt of a polymer. "Grafting-to" reactions show limitations due to steric hindrance between polymer chains, which makes them difficult to bond to the surface at short intermolecular distances [46].

"Grafting-from" reactions are an example of covalent attachment of a polymer in situ on the solid substrate. This method requires an initiator of polymerization, such as azo-, peroxide- or photo-initiators coupled to the surface of the substrate. These reactions are preferable because they enable the control of the grafting density and film thickness. Advanced methods of performing "grafting-from" reactions comprise living radical polymerization strategies: atom transfer radical polymerization (ATRP), reversible addition-fragmentation chain transfer polymerization (RAFT), and nitroxide-mediated polymerization (NMP). Radical-based "grafting-from" reactions are commonly used because of their compatibility with aquatic and organic media since they have a high tolerance

toward various functional groups [41,43]. The "grafting-from" reactions are often initiated by UV or visible light [43,46–50].

As opposed to covalent functionalization, where it comes to the formation of strong permanent bonds, non-covalent functionalization relies on dipole–dipole, van der Waals and other interactions such as hydrogen bonds. For carbon, non-covalent interactions occur between unhybridized π-orbitals of $sp^2$ carbon atoms and cations, anions or molecular species.

Gases especially tend to form non-covalent bonds with carbon as they are characterized by weak molecular interaction in their aggregation state [51]. Non-covalent attachment of molecular species can also proceed through π-π stacking, hydrophobic/hydrophilic interactions, or van der Waals interactions, thus still preserving intrinsic electronic properties of graphitic carbons [31,52]. Non-covalent functionalization of carbon is preferred when there is a demand for preserving not only intrinsic, mechanical and electronical properties, but also bulk and surface properties. In the comprehensive review by Zhou et al. [53], where different non-covalent functionalization strategies are discussed, it is highlighted that this approach enhances the bio-affinity of carbon nanotubes (CNTs).

The second example of a successful non-covalent functionalization of CNTs is a work by Panchakarla and Govindaraj [54]. This group performed both covalent and non-covalent functionalization of CNTs which was achieved through π-π interactions between the benzenoid groups in CNTs and three different polymers (polyethene glycol (PEG), polyoxyethylene(40)nonyl-phenyl ether (IGPAL) and 1-pyrenebutanoic acid succinimidyl ester (PYBS)), which resulted in a high solubility of CNTs even at a low level of functionalization. Other examples are related to the "wrapping" of carbon particles with poly(vinyl alcohol) [55], poly(sodium 4-styrenesulfonate) [56], surfactants [57] and other polymers [58].

### 3.1. Oxidation of Carbon Surface

Surface oxidation is a commonly used approach for introducing precursor oxygen groups on the carbon surface. It allows for an easy surface modification with desired functional entities for the subsequent steps of tailoring the carbon surface. Wet chemical oxidation can be performed in various solvents (and their mixtures) such as nitric acid ($HNO_3$), sulphuric acid ($H_2SO_4$) hydrogen peroxide ($H_2O_2$), ammonium persulfate (($NH_4)_2$ $S_2O_8$), hypochlorites (NaOCl) bichromates ($K_2Cr_2O_7$) and many more.

One of the most common methods for oxidizing a carbon surface is the procedure developed by Hummers and Offeman in 1958 [59]. This oxidation method was developed from previously investigated methods [60] which showed various drawbacks such as the formation of toxic gases and long reaction times. Over the years, the original Hummers method and its more recent variations [61–70] have become a confident way of producing graphene layers by exfoliating graphite via inter-layer functionalization and functionalization of carbon surfaces. Most of the frequently used oxidation methods are not catalysed. The oxidation of the carbon surface occurs in direct contact between carbon and oxidizing agents. In the original reaction of Hummers and Offeman, $NaNO_2$ can be considered as a catalyst in oxidation and possible catalytic characteristics of $NaNO_2$ have been reported elsewhere [71]. In finding a novel method for the modification of an inert carbon surface, there is not only a demand for reducing costs and time of production but also lowering or, in the best case, avoiding harmful effects on the environment. The novel methods of oxidation are often upgraded to commonly used ones and they are meeting today's demands: low energy consumption, low production costs and an environmentally benign approach. Positive examples of upgraded wet chemical oxidation are hydrothermal and solvothermal methods. Moreover, dry methods using continuous oxidation with controlled plasma are known. This type of oxidation can be achieved by dielectric barrier discharges (corona discharges), microwave plasmas, and so on [72]. In this chapter, frequently used approaches of oxidation are described, together with a review of the underlying reaction mechanisms. We want to emphasize that the examples of oxidation methods are refer-

ring to different carbon types, but also can be employed in the surface modification of pyrolytic carbon.

### 3.1.1. Structural Evolution of Surface Oxygen Groups

Many researchers have investigated the interaction between solid carbon and oxygen in, e.g., combustion or gasification [73]. Oxidation of carbon is nothing else but a chemical reaction where oxygen entities are introduced at active sites to lower the energy of carbon. Active sites are usually defects in carbons, such as in-place vacancy defects, edges of aromatic rings or aliphatic residues. These edged active sites are characteristic of materials with larger pore channels (macro- and mesoporous, due to geometrical hindrances) [74]. It is known that a successful functionalization of the carbon surface depends on availability of the surface of the carbon (area where reactions occur). For example, if there are, e.g., no formed agglomerates and/or bundles, hydroxyl and epoxy groups are usually formed on the basal plane of the aromatic structure. On the other side, carbonyl and carboxyl groups are found on the edges of the aromatic planes [63,75,76].

Du et al. [77] proposed a two-step mechanism, and it includes different active sites: type A (very reactive and firstly saturated) and type B (less reactive, slow oxygen adsorption sites). This mechanism suggests that oxidation is the chemisorption of $O_2$ molecules which are not homogeneously dispersed on the carbon surface but attached as clusters of molecules. The functional groups formed on the carbon surface are going through an evolution that is affected by the type of active sites and the reaction parameters. Since a typical carbon material has a complex surface structure, various oxygen species will be created because of differences in $O_2$ adsorption.

Lear et al. and Hynes [78,79] proposed a mechanism where the chemisorption of $O_2$ firstly occurs on the active site, and forms metastable carbon–oxygen species. Further, it undergoes regrouping in an oxygen-rich environment or metastable species can be rearranged between themselves into energetically favourable forms with $CO_2$ or CO as byproducts. The last step is oxygen desorption, which occurs at higher temperatures and releases gaseous products ($CO_2$ or CO).

The edges of graphite planes are most likely to be attacked since there are no steric hindrances, so it makes them accessible for attaching to oxygen groups (Figure 1). Graphitic planes tend to adsorb neighbouring molecules to keep the aromaticity of ring structures [80]. After chemisorption, disrupted $sp^2$-hybridized atoms are converted into $sp^3$-hybridized atoms and the following step is reorganization into the most energetically favourable state. In the early stages of oxidation, chemisorption begins with the attachment of oxygen atoms over two neighbouring carbon atoms, forming reactive and metastable epoxy groups. Epoxy groups are the least stable and go through a reaction sequence over unstable complexes of peroxide, hydroperoxide and hydroxyl species [81]. During longer exposition to an oxidizing atmosphere, there is a reduction in unstable groups' results in the formation of stable carbonyl-containing species. This includes semiquinones, lactones and carboxylic groups, which are generated in the presence of oxygen and hydroxyl radicals at elevated temperatures and display characteristic oxygen-containing groups at carbon surfaces [82].

### 3.1.2. Chemical (Wet) Oxidation and Overview of Reaction Mechanisms
Oxidation with $HNO_3$, $H_2O_2$ and $(NH_4)_2S_2O_8$

Activated carbon was investigated for the adsorption of various ions from wastewater by Wu et al. [83]. Oxidation was performed with $HNO_3$, $(NH_4)_2S_2O_8$ and $H_2O_2$. $HNO_3$ is mostly used because of ease in controlling reaction parameters, but it is very strong and causes deterioration in the structure of carbon and produces highly toxic $NO_X$ gases. $H_2O_2$ is also effective in generating oxygen groups, but it is toxic and also damages the structure of carbon. During oxidation with $(NH_4)_2S_2O_8$, oxygen species are generated without affecting the structure. The authors investigated the pore evolution, mesostructural stability of the carbon and surface functionalization. The authors confirmed excellent mesostructural stability of carbon under strong oxidation conditions (with $HNO_3$) as well

as drastic changes in carbon pore structure due to deterioration during oxidation processes. It is shown that structural regularity of carbon can be preserved for powerful agents at lower temperatures for 12 h. In weaker oxidizing agents, such as $(NH_4)_2S_2O_8$, the structure can be preserved at lower concentrations (also at a higher temperature) for 72 h. It was confirmed that the creation of acidic groups, such as –COOH, facilitates the dispersion of carbon in aqueous environments and gives the possibility to adsorb metal ions, drugs and dyes.

**Figure 1.** Oxygen-containing surface groups (*a colour version of this figure can be viewed online*).

As reported by Houshmand et al. [84] the concentration of $HNO_3$ is the prime factor in the modification and the texture of the carbon surface. Since activated carbons are usually used as adsorbent, catalyst or catalyst support, the modification of textural characteristics and surface chemistry are the primary targets in tailoring desired surfaces. The surface chemistry of microporous activated carbon was changed in nitric acid solution by varying the concentration, time and temperature of the reaction. An increase in the number of functional groups decreased the amount of specific surface area. The authors concluded that by changing the concentration of $HNO_3$, minor changes occur in surface texture. This confirms that modification with nitric acid is a convenient approach towards the controllable oxidation of surfaces. The authors also developed different models that describe changes in BET surface area and oxygen index. The oxygen index is the number of oxygen surface groups available on the sample (calculated from the area below the temperature-programmed desorption (TPD) curve). It was shown that the BET surface area can be predicted with a linear model and the oxygen index can be described with two-factor interaction. Both of them strongly depend on acid concentration, temperature and time. It was also shown that surface area, total pore volume and micropore volume were negatively affected even under mild experimental conditions, which proves that further oxidation leaves a negative effect on the textural characteristics. However, since the surface chemistry is more influenced, the creation of the oxygen groups can be easily controlled by adjusting the oxidation parameters.

Su et al. [32] oxidized activated carbon in $HNO_3$ of varying concentrations to obtain a more polar material and changes in pore size distribution, which strongly influences the adsorption of heavy metals in wastewater treatment. The fundamental goals were to modify the surface of activated carbon with different concentrations of $HNO_3$ and to examine the adsorption capacity of the carbon. The results showed that during $HNO_3$ treatment at higher concentration, there is a possibility of integrating $-NO_2$ groups into the carbon structure. The increasing negative charge on carbon structure is caused by a higher amount

of oxygen functionalities (e.g., –COOH), which affects adsorption and interaction with the solvent. By comparing different oxidized samples, a dependency on concentration and time was discussed. Surface oxidation with the highest acid concentration (20% $HNO_3$) did not give the best results, because of the negative charge on the surface. It is worth mentioning that $Ni^{2+}$ and $H^+$ are competing on the adsorption sites which causes a repulsion effect between these cations. That was confirmed by the finding that the highest amount of adsorbed $Ni^{2+}$ appears on activated carbon treated with lower acid concentration (15% $HNO_3$). Therefore, the latter material was selected for the further study because of its relatively high pore volume and surface area. It was proved that modified carbon with acidic surface groups provides a higher adsorption efficiency. The authors also proposed a kinetic model (pseudo-second-order) that fitted their data very well. It was concluded that $Ni^{2+}$ adsorption was affected not only by physical adsorption over highly developed micropores, but also over ion exchange between $Ni^{2+}$ and the functional groups introduced onto the surface.

Oxidation of the carbon surface is the preferred surface modification of carbon, and wet oxidation has been described by many authors. In a paper by Morreno-Castilla et al. [85], a series of activated carbons (obtained from olive kernels) were oxidized with different oxidizing agents, in particular $HNO_3$, and aqueous solutions of $(NH_4)_2S_2O_8$ and $H_2O_2$. The authors investigated changes in the surface chemistry of activated carbons after treatment with these oxidizing agents since each one of them produces different surface oxides: acidic, basic and neutral groups. It is known that $(NH_4)_2S_2O_8$ produces stronger acid groups than $HNO_3$ and does not significantly change the surface area and the pore structure of activated carbon [86,87]. It was shown that treatment with $(NH_4)_2S_2O_8$ gave the lowest amount of oxygen surface groups compared to other agents. The binding of the oxygen groups strongly depends on the availability of active sites on carbon. Due to this, it is important to distinguish two different available surfaces on activated carbon: the *external surface* (geometric surface, i.e., the boundary of a carbon) and the *internal surface* (porosity). According to reference [85], surface treatment with $(NH_4)_2S_2O_8$ and with $H_2O_2$ fixed the largest amount of oxygen groups on the external surface. Oxidation with $HNO_3$ influenced the internal surface and, along with $H_2O_2$, caused changes in macropore volume: the treatment with $HNO_3$ decreased the macropore volume, while treatment with $H_2O_2$ increased the macropore volume. The authors observed that rearrangement of oxygen groups differs between internal and external surfaces; the internal surface was more basic than the external surface. It is confirmed that oxidation with stronger agents, such as $HNO_3$, increases the amount of acidic functional groups on the surface, which affects the structure of the carbon skeleton as well.

Reaction Mechanisms of Carbon Oxidation by $HNO_3$ and $H_2O_2$

As already explained, the formation of oxygen groups is likely to happen on the most reactive carbon atoms [82]. Active sites may arise from various causes, such as foreign inorganic matter (ions), impurity of material or defects of the carbon surface [73]. The amorphous phase in carbon is very prone to oxidation, tending to produce carboxylic species on its surface. The carbon surface, e.g., in turbostratic carbons contains defects (such as carbon pentagons and heptagons) that cause strain and, therefore, a tendency to become involved in oxidation. This paragraph is dealing with the most common oxidizing agents, $HNO_3$ and $H_2O_2$, in order to give an insight in the mechanisms that are occurring on the surface of (different) carbon materials.

The paper by Kanai et al. [88] considers the basic approach to oxidation of CNTs in $HNO_3$ through physisorption, chemisorption and ether- and epoxy-incorporating configuration (Figure 2). Each of them occurs simultaneously. The goal of this research was to enhance the desired process by lowering secondary interactions. Their work showed that the oxidation state highly affects stable chemical states and arrangements of the system. The authors observed two different mechanisms. In the first mechanism, oxygen entities are formed as a result of physisorption caused by local charge transfer. The second mechanism

is based on chemisorption, which becomes more favourable when the carbon oxidation state is increased. Compared to physisorption, chemisorption is less energetically favoured due to defects of the graphene layer caused by changes in the hybridization state of carbon atoms (to sp$^3$). Spontaneous dissociation and oxygen incorporation are the least favourable competing reactions. The authors showed that epoxy and ether structures are not energetically favourable and that they are less stable than chemisorbed moieties because of structural tension. The researchers concluded that the difference between physisorbed and chemisorbed oxidized states decreases with a higher amount of nitro groups being present.

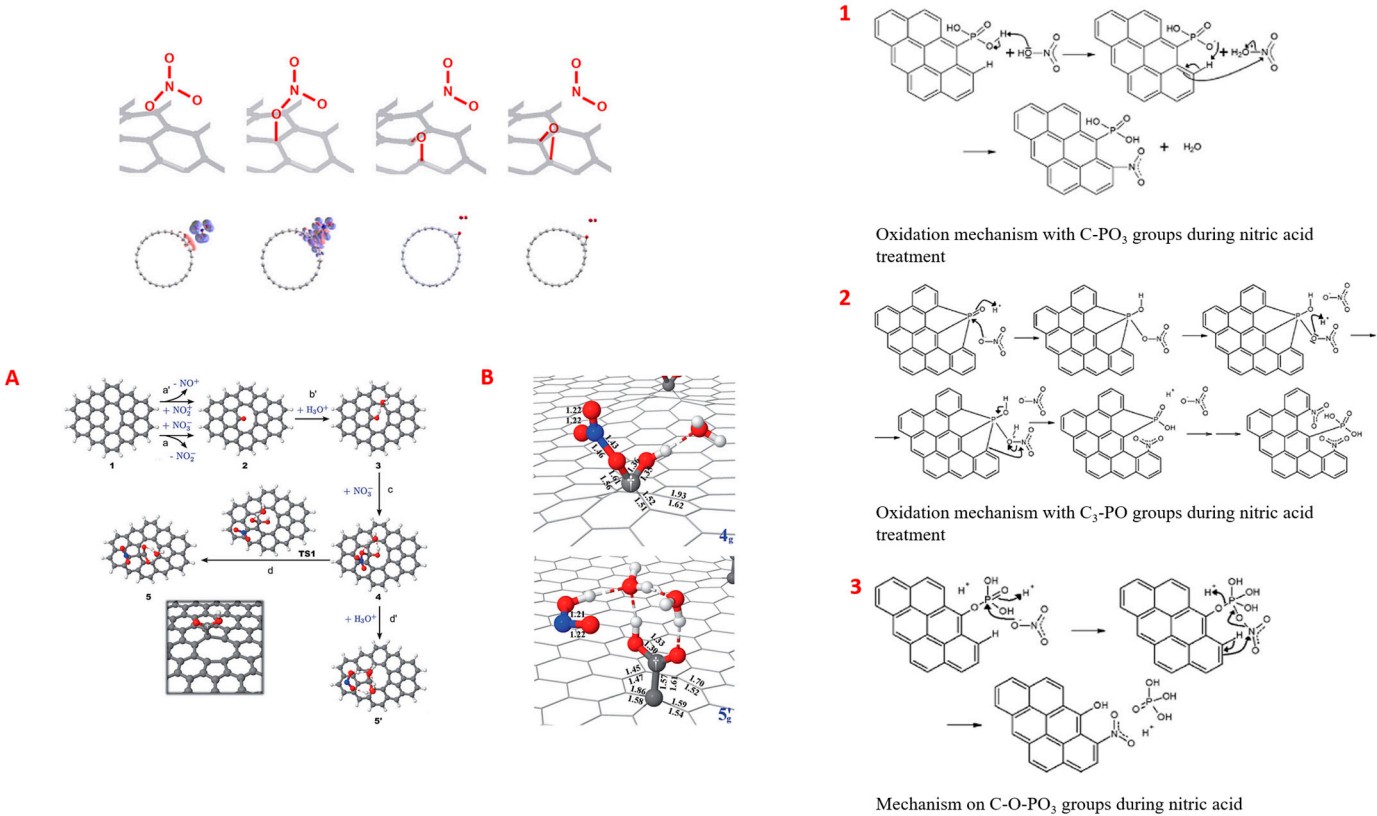

**Figure 2.** Chemical states of oxidated carbon: (from left to right) physisorbed, chemi-sorbed, ether- and epoxy configuration, copyright, 2010, Reproduced with permission from American Physical Society [88] (top); the nitric acid oxidation (**A**) (a–d, d') and (**B**), copyright, 2016, Reproduced with permission from Wiley-VCH Verlag GmbH & Co. KGaA [82] (middle); and the oxidation of different phosphorus groups (**1–3**) copyright, 2016, Reproduced with permission from Elsevier [89] (down) (*a colour version of this figure can be viewed online*).

Ternero-Hidalgo et al. [89] performed the oxidation of carbon with HNO$_3$ in combination with alternate oxidation in air. The principal aim was to investigate the influence of phosphorus groups on the functionalization of activated carbon. The carbon was firstly treated with phosphoric acid, which forms stable phosphorus complexes. The results showed that upon further treatment with HNO$_3$, nitrogen groups are grafted (e.g., nitro groups, –NO$_2$). Different mechanisms of oxidation were described. The authors assumed that all three mechanisms are involving different phosphorus groups, see Figure 2. In the first mechanism (1), the nitration mechanism involves a reaction between C–PO$_3$ and HNO$_3$. A proposed mechanism suggests that the phosphorus groups are acidic sites and act as catalysts for aromatic nitration.

In this case, highly reactive nitronium ions, NO$_2^+$ are formed after protonation of nitric acid. These moieties are active species in electrophilic aromatic substitution (EAS).

The $NO_2^+$ ion is not very stable; it quickly disappears and forms nitro groups (–$NO_2$) on the aromatic structure.

The second mechanism (2) includes nitration that occurs on $C_3$-PO groups. Here, phosphorus complexes (C–$PO_3$/ $C_2$–$PO_2$) with lower oxidation states are created from the oxidation of $C_3$–PO groups. On the created phosphorus complexes, P=O is protonated after, followed by aromatic nitration. The following step includes bonding of nitrates onto the phosphorus atom to produce $NO_2^+$ ions and –$NO_2$ groups, after which the P=O moieties are recovered.

The third mechanism (3) proposes that nitration occurs in the presence of C–O–$PO_3$ moieties. Here, oxygen in P=O is protonated and –$NO_2$ is bonded to C–O–P complex. A further step includes breaking of the O-P bond and formation of the phenol. Here, phosphoric acid is a byproduct (released into solution) and -$NO_2$ groups are formed. Due to the fact that a fraction of phosphorus is removed, this mechanism is the least favoured. It is proposed that all named mechanisms take place simultaneously and all of them result in the formation of nitro groups.

A four-step oxidation mechanism of carbon in $HNO_3$ is presented by Gerber et al. [82] and it is based on a theoretical approach. These researchers observed oxidation at vacancies (i.e., defects), a transformation of phenol to carbonyl groups, formation of intermediates and formation of carboxylic groups as the fundamental reactions that occur during the formation of oxygen functionalities (Figure 2). This mechanism considers the lowest energy scenario, both from thermodynamic and kinetic perspectives, and it highlights the enlargement of initial vacancies. The proposed mechanism includes the fast formation of –COOH groups on the structural defects from phenol groups on the other sites. Based on their calculations, the reactions would start with a fast carbonylation of the vacancies, followed by phenol formation. First, nitric acid is subjected to self-dehydration and oxidation, producing reactive ionic species ($NO_2^+$, $H_3O^+$ and $NO_3^-$), where the $NO_2^+$ cation participates in the most thermodynamically favoured oxidation processes. The attack (a) of an $NO_3^-$ anion on a carbon atom (1) produces an oxygenated group (2) and nitrite anion as a by-product. In the second step (b) the protonation of (2) leads to a hydrated functional compound (3). The hydrated functional compound now forms hydrogen bonding with oxygen in grafted $NO_3^-$, species which result in formation of nitroso group. This intermediate undergoes subsequent dissociation where a carboxylic group can be formed. There are two ways to form a –COOH group at defective sites: overextension of the existing vacancies or interaction of the reactive ionic species with grafted nitro groups. Carbon atoms can attract intermediate complexes, which causes a creation of a five-membered ring closure. The second way is explained by the interaction of $H_3O^+$ with grafted $NO_3^-$ and induces aligned oxygen transfer into the nitro-hemiacetal carbon atom, which leads to a surface bound carboxylic acid. Overall, this mechanism is based on the most efficient pathways leading to the formation of –COOH units on the carbon surface. It includes formation of the oxygen functionalities on the vacancies. The authors showed that formation of the oxygen groups can be performed at room temperature even though all experiments were conducted on elevated temperatures and that further formation of carboxylated species is based on the migration and coalescence of the present vacancies that would lead to their enlargement.

In the paper by Westbroek and Temmerman [39], the mechanism of hydrogen peroxide oxidation at glassy carbon electrodes is described. The authors examined the voltammetric behaviour of $H_2O_2$ in an alkaline solution. Since the voltammetric current strongly depends on the pH and the temperature, the proposed mechanisms give an insight in the formation of the various species. Therefore, two mechanisms are proposed (A and B) which are in competition and where reaction orders depend on $OH^-$ concentration. Mechanism A occurs at low $H_2O_2$ concentrations, while mechanism B is characteristic of higher $H_2O_2$ concentrations. The paper presents the mechanism of $H_2O_2$ degradation for sensor systems, but in this case, it gives us an overview of what is happening with hydrogen peroxide in contact with carbon. The authors used high density glassy carbon to avoid the risk of absorption of oxygen that is produced during the reaction. The overall mechanism

starts with the decomposition of $H_2O_2$ (to $HO_2^-$ and $OH^-$) and ends with the formation of molecular oxygen. In the first steps of both mechanisms, $HO_2^-$ is adsorbed on the carbon surface and, since it is highly unstable, it decomposes into $O^-$ and $OH\bullet$. In the second step, $OH\bullet$ reacts with another $OH^-$ and produces $O^-$ and $H_2O$. The two mechanisms can be distinguished by the reaction products: in reaction mechanism (A) with $O^-$ as a starting point, molecular oxygen $O_2^-$ can be formed on the carbon surface, while in reaction mechanism (B), $OH^-$ is also found besides $O_2$. In the last step, the $O_2^-$ formed on the surface is oxidized to gaseous $O_2$. It was observed that the overall reaction mainly depends on the $H_2O_2$ concentration, which causes varying reaction orders. The authors used the proposed mechanisms to describe the kinetics for each of them. This can be applied in amperometric measurements of unknown peroxide concentration from a known signal, pH value and temperature. Although this mechanism is not explaining what is happening to the carbon itself, knowing the oxidation mechanism of hydrogen peroxide is important to explain the creation of oxygen functionalities on the carbon surface. Furthermore, an additional insight into hydrogen peroxide oxidation can be found in the paper by Sengupta et al. [90]. The researchers presented a list of 27 reactions in the thermal decomposition of $H_2O_2$ since oxidation of carbon with hydrogen peroxide can be conducted without the presence of catalysts.

In the already mentioned paper by Du et al. [77], the reactions of carbon and oxygen were studied to understand reaction mechanisms in terms of elementary processes of adsorption, complex formation, rearrangement and desorption. Their starting point was postulated by Ahmed et al. [91] where carbon oxidation occurs also as a two-site mechanism with continuous site energy distribution. If combined with the previously explained mechanism of $H_2O_2$ oxidation [39], the following could also be considered as a mechanism of oxygen group formation with hydrogen peroxide. If the mechanism proposed by Du et al. [77] is based on the assumption that different carbon groups are formed during chemisorption of $O_2$, then the following can be described for the sequential formation and evolution of acidic and basic groups. It all starts with the chemisorption that takes place on two sites that differ in energy: on a highly reactive site in terms of $O_2$ adsorption (A) and subsequent reactions on site (B) (*vide supra*, Section 3.1.1). In the first steps, $O_2$ is adsorbed on the carbon surface and thus complex species on both sites are formed. If it is assumed that both sites (A and B) are evenly arranged on the carbon surface, it can be said that all probable reactions are occurring simultaneously. The oxygen is adsorbed and forms a complex structure ($C–O_2$) which can be rearranged and, therefore, forms two ($C–O$) complexes. This could be linked with the assumption that an epoxy bridge is formed over two neighbouring carbon sites. The acidic groups can be formed by the interaction of neighbouring carbon sites, ($C–O$) complexes and $O_2$ which can result in the formation of carbonyl and carboxyl groups. In the oxidation of activated carbons, the structure is affected by the cleavage of hydrogen peroxide and oxygen production, along with the functionalization of their surface. It could be proposed that $H_2O_2$ in reaction with a benzenoid ring in the graphene layer firstly results in the formation of an epoxide group. After the epoxide ring has been formed, it undergoes a rearrangement to a phenolic group because of its high instability. From the phenolic group as the starting point, carbonyl and carboxyl groups can be further formed (Figure 3).

To conclude, $H_2O_2$ and $HNO_3$ are commonly known as strong oxidizing agents, whilst the mechanisms of oxidation of carbon are not completely known. Both of them are widely used since the oxidation of carbon can be easily modified just by modifying reaction parameters. It is a proven method of wet oxidation to achieve polar carbon surfaces. The presented examples are just a brief overview of the reactions that proceed in the system. Therefore, it can give us only a limited picture about the formation of the oxygen functionalities on the carbon surface.

**Figure 3.** Formation of oxygen containing groups (oxidation of carbon with peroxide solution) (*a colour version of this figure can be viewed online*).

Hydrothermal and Solvothermal Oxidation Methods

Hydrothermal and solvothermal methods are important in the synthesis of materials and in the treatment of wastes by mimicking geothermal and hydrothermal processes. The hydrothermal method applies to chemical reactions with water at temperatures higher than the boiling point of the water. The solvothermal method is performed at high temperatures (100–1000 °C) and high pressures (1–100 MPa) in non-aqueous solvents [92]. Hydrothermal and solvothermal methods are carried out in autoclaves and are used when there is a need for increasing reaction temperatures [93]. A pure hydrothermal approach can also be performed with small quantities of commonly used oxidizing agents (e.g., $HNO_3$ and $H_2O_2$) added to water. The results showed that the oxidative hydrothermal approach has great potential, which was proven by the amount and types of oxygen containing groups. The authors also showed that higher temperatures cause the disappearance and also the re-formation of surface functionalities.

These chemistries are preferably carried out to produce complex inorganic materials by accelerating, e.g., the ionization of the oxidizing agent by controlling temperature and pressure in safely sealed containers. Under such conditions, a reaction between a solid, and the oxidizing agents occurs on the surface by fast exchanging ions and molecules. Here, colloidal systems favour chemical and physical homogeneity. Significant advances in using these technologies are in overcoming issues such as high energy consumption, production of toxic gases, safe handling and convenient adjustment of pressure, pH and temperature.

In the work by Silva et al. [94], hydrothermal synthesis was used for the precise modification of carbon xerogel at a low concentration of $HNO_3$ in water. Based on these results, it is of high importance to choose the right ratio between carbon and oxidizing agent as this affects the number of oxygen functionalities introduced onto the carbon surface. In this way, carbons oxidized in $HNO_3$ can be used as catalysts and adsorbents. The authors concluded that the level of oxygen functionalities depends on the treatment conditions, i.e., acid concentration, temperature and amount of carbon. The texture of carbon was

not significantly affected and it was also found that treatment with higher concentrations of $HNO_3$ leads to an increased number of functionalities. The same research group [95] also examined the controlled oxidation of other carbon materials, e.g., carbon nanotubes, by hydrothermal oxidation in $HNO_3$. The authors performed experiments with the goal of establishing suitable mathematical models by which a functionalization of CNTs is predicted for a whole range of ratios $HNO_3$/CNTs. It was concluded that type and amount of oxygen functionalities also depend on the morphological structure of the carbon material. Further work of these authors was devoted to the hydrothermal oxidation of CNTs with other oxidizing agents, such as $H_2SO_4$ and $(NH_4)_2S_2O_8$, in order to produce bucky paper (thin sheet of aggregated CNTs) [96]. The results showed that, for the named experiment, attached functionalities depend on the length of graphene layers of the CNTs. According to the results, CNTs treated with a mixture of $H_2SO_4$/$(NH_4)_2S_2O_8$ show more acidic character because of more sulphur groups present at the surface. Compared to samples treated with $HNO_3$, more oxygen groups are created. In the examined morphologies, a decreasing trend of functional groups is observed when going from multi-walled to single-walled CNTs. This is due to the number and length of graphene layers, and thicker and shorter CNTs offer more reactive sites than thinner and longer CNTs. It was concluded that both $HNO_3$ and $H_2SO_4$/$(NH_4)_2S_2O_8$ act as efficient oxidation agents for the precise control of surface functionalization via a hydrothermal method.

Wang et al. [97] functionalized CNTs to produce CNT/epoxy composites. The authors examined the influence of ozone ($O_3$) as the oxidizing reagent. In this work, CNTs were treated with mixtures of different oxidizing agents: $HNO_3$, $O_3$ and $H_2O$, ($H_2O$/$O_3$) and also with $H_2O_2$ added ($H_2O$/$O_3$/$H_2O_2$). Oxidation with $H_2O$/$O_3$/$H_2O_2$ showed the best results and it was optimized to be suitable for industrial-scale production. It was observed that there was an improved chemical reaction and the compatibility between the polymer matrix (epoxy resin) and carbon filler. The group successfully examined a scaled-up production with this process by which environment pollutions (caused by oxidation with $HNO_3$) were avoided. To examine hydrothermal oxidation of carbons to produce epoxy composites, another work of this research group [98] was related to performing experiments in an autoclave. The authors compared two different hydrothermal methods: one was reacting CNTs with pure water at different temperatures, and the other was oxidation of CNTs with $HNO_3$. As a result, the created carboxylic and phenolic groups on the carbon contributed almost twice the amount of oxygen that was found for the pristine (raw) carbon samples. Although the best oxidation performance was still achieved with $HNO_3$, oxidation by the hydrothermal method (using pure water) at an elevated temperature resulted in relatively high amounts of oxygen species at the carbon surface. The authors concluded that oxidation with hot water did not cause any damage to the carbon structure which is in contrast to the treatment with $HNO_3$. This is a proof that hot water treatment is a promising method in "green chemistry" applications.

Ang et al. [99] carried out the oxidation of carbon in an autoclave with water and oxygen at elevated temperatures. For this hydrothermal oxidation, the commonly used wet oxidation was combined with elements of the dry oxidation method. It provides a unique synergy and is presented as a low-toxic and environmentally friendly method. These hydrothermal treatments were performed in the temperature range from 120 °C to 350 °C, which is below the critical point of water (374 °C). Under these conditions, water acts as an acidic (or basic) precursor that influences chemical reaction rates [22,95].

Summing up, hydrothermal and solvothermal oxidation of carbon are promising methods to create a sustainable way of modifying carbon surfaces. By performing reactions under safer conditions and with easier control of reaction parameters, hydrothermal and solvothermal reactions are suitable upgrades of commonly used wet oxidation methods. According to the literature, these methods provide the possibility of using clean water as reaction medium, as higher temperatures can be reached and oxidising gases can be added. Overall, these methods are also promising green chemistry approaches whose benefits can be implemented in various fields and on an industrial scale.

### 3.1.3. Physical (Dry) Oxidation and Overview of Reaction Mechanisms

Dry oxidation of carbon implies a permanent or semi-permanent functionalization of the surface in the presence of ionized gases. One of the most common methods for dry oxidation is ozone treatment [100–106]. Oxidation treatments with ozone, in combination with other agents, are usually referred to as "advanced oxidation processes" (AOPs). In general, AOPs are all the methods where OH radicals are formed, such as in Fenton reactions (cleavage of $H_2O_2$ with $Fe^{2+}$), UV reactions (cleavage of $H_2O_2$ under UV light) and ozonation (reaction between $O_3$ and $H_2O$) [100]. However, these processes are mostly used in the treatment of organic compounds in wastewater.

In the context of oxidation with excited gasses, an oxidation that takes place under atmospheric conditions, by ionization of air, is termed as "corona discharge oxidation". When oxidation is achieved at very low pressure (vacuum) and in the presence of specific gases such as $O_2$, $N_2$ and Ar, it is termed "plasma oxidation". Sometimes, both corona discharge and plasma oxidation are termed as "plasma reactions". Although corona treatment is mainly used for treating polymers [107–109], similar methods—such as plasma oxidation—are well studied for the treatment of carbons. In corona oxidation, a high voltage and high frequency electrical discharge is applied [110], where gases such as oxygen and ozone are ionized at atmospheric pressure.

#### Hydrophobic Recovery

The main aim of treating different surfaces with, e.g., oxygen and nitrogen plasma is—in most cases—to render them more hydrophilic. Oxidative plasma and corona treatments increase the surface energy and thus the wettability of a surface and is mostly used for activation and modification of polymers. Although it presents an attractive alternative to the commonly used chemical functionalization process, functionalization via plasma suffers from one significant drawback: hydrophobic recovery [72,111–114]. This means that the functionalities which are formed at the surface via the plasma (or corona discharge) treatment are not permanent [115–121]. As a result, the hydrophilic nature of a surface vanishes within hours or days of storage under ambient conditions. Hydrophobic recovery is commonly explained for polymers [122] treated with plasma (or corona discharge), but it can be also applied to other materials, e.g., carbon.

In general, surface functionalization with plasma is affected by various process parameters along with characteristics of a material itself (e.g., natural aging of the material, surface roughness or influence of the present contaminants). Factors that are responsible for hydrophobic recovery can be summarized as [118]:

- Overturn of the polar groups at the (polymer) surface, i.e., the created hydrophilic groups re-orientate away from the surface;
- Migration of the created polar moieties from the surface to the bulk (outside-in);
- Migration of the untreated moieties through the bulk matrix to the surface (inside out);
- The loss of volatile, e.g., oxygen rich species (and other polar functionalities) to the atmosphere;
- A change in surface roughness.

The overturn and migration of newly created polar groups away from the surface is explained by the fact that air is a non-polar gas and the contact of polar surface groups with air is, therefore, energetically unfavourable.

The common approach for checking the stability (or instability) of a surface is contact angle measurement and FT-IR and XPS spectroscopy. With contact angle measurements, an increase in the contact angle of water indicates that plasma created functionalities are removed from the surface due to one or the sum of the listed effects, and that the surface becomes less polar again. By performing spectroscopic analysis, the loss of specific bonds (FT-IR) and changes in the elemental composition (XPS) can be observed as a result of hydrophobic recovery. Hydrophobic recovery is also linked with the natural aging of the material (and its surface). A commonly used approach to overcome this problem is

prolongation of exposure to plasma and/or changing its parameters, and storage of the treated sample under a polar atmosphere.

However, it is known that longer exposure to plasma, continuous treatment at higher power values or exposure to stronger agents can exert a negative influence on the treated material, such as creation of (sometimes) visually seen defects of treated surface, vacancies on the micro level, or formation of debris. The effect of hydrophobic recovery needs to be regarded when performing surface functionalization followed by storage of the modified material. This also refers to the finding of a possible solution for preserving a material's surface stability.

Reaction Mechanisms of Plasma-Assisted Carbon Oxidation

In this section, the reaction mechanisms of carbon oxidation under plasma conditions are dealt with. Both classical plasma processes under vacuum and also plasma treatments under atmospheric conditions (corona treatments) are discussed.

In plasma oxidation, high voltage is applied in a gas atmosphere (a few mbar), and ionized species such as $O^-$, $OH^- \bullet$ and $O_3$ are formed when oxygen molecules are excited by argon ions (and/or $NO_2^-$ if nitrogen is present) [5]. Plasma reactions can also be used to remove particulate matter (PM) in exhaust gases (also termed "soot"). The review by Shin et al. [23] describes reactions mechanism, patents and future developments. PMs are produced during the combustion of fuels and appear as ashes (from lubricants and oils) and sulphur compounds (from diesel and water). PMs are assumed to be carbon materials with graphene-like structures. The mechanism explained by Ju et al. [6] can only be applied if the plasma oxidation is combined with combustion in motor systems, because plasma enables controlled ignition and flame stabilization in high-speed propulsion systems.

The paper provides a mechanism of carbon oxidation in plasma discharges at low temperatures and describes four different enhancements of plasma-assisted combustion: thermal, kinetic, diffusion and convective transport of excited species, which occur simultaneously in the system. In studying the kinetics of oxidation in plasma discharge, Ju et al. [6] presented plasma-assisted combustion where radical species generated by the plasma speed up the decomposition of fuels in different temperature ranges. This paper gives a picture of how plasma influences the processes in the presence of oxygen.

Applying this mechanism to the oxidation of solid carbon, Shin et al. [5] and Yao et al. [123] explained the reactions at the carbon surface. Plasma discharge (dry oxidation) of carbon-like particulate matter proceeds in a so-called "unzipping mechanism", where the incorporation of oxygen groups initiates cracks in the graphene layer. Yao et al. [123] proposed a two-site mechanism (similar to the ones explained in Sections 3.1.1 and 3.1.2), but related to dry oxidation. In this mechanism, the plasma discharge produces atomic species from molecular oxygen. Not all molecules are fully excited and some of them are adsorbed on the carbon surface at sites that differ in energy (A, B). The first pathway is the adsorption of $O_2$ which interacts with a carbon atom and forms an oxygenated group. In combustion, the oxygenated group reacts with $O_2$ to form $CO_2$. The second pathway is the formation of an epoxide group between one oxygenated carbon atom and its neighbouring carbon atom. The transition complex can produce CO, or it can form an oxygen group on one carbon atom and another free carbon site.

Another mechanism of carbon oxidation in plasma discharge can be applied when PMs are removed from the exhaust gases of diesel engines. It is presented by Lu et al. [111] for the catalytic oxidation of graphite-like carbon. Based on their results, the group proposed the following mechanism (Figure 4) of catalyst performance in oxidation (M stands for metal catalyst):

- $O_2$ is excited and decomposed into two O atoms where one directly reacts with the catalyst surface and forms M-O, and the other reacts with $O_2$ and forms $O_3$;
- $O_3$ reacts with the catalyst and forms another M-O unit, $O_2$ and/or intermediate species: $M-O_3$ and/or $M^+-O_3$;

- Since the aluminium catalyst is present in the system, it is the source of residual $H_2O$ (M-$H_2O$) which can form M-OH and a bicarbonate complex M-O($CO_2$);
- The bicarbonate complex reacts with the oxygenated catalyst M-O and forms M-O($CO_2$).

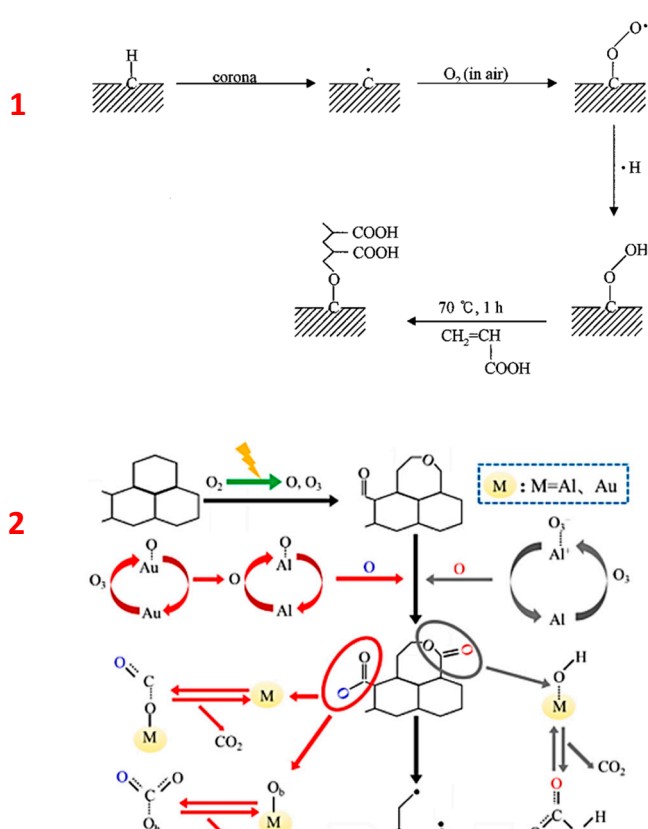

**Figure 4.** 1-Dry oxidation mechanisms (corona discharge oxidation and plasma oxidation): 1-Mechanism of –COOH group formation grafted surfaces by corona discharge with following graft copolymerization, 2016, Reproduced with permission from Elsevier [108] (**top**); 3-Oxidation of carbon via plasma oxidation in presence of a metal catalyst, 2020, Reproduced with permission from Elsevier [111] (**down right**) (*a colour version of this figure can be viewed online*).

The bicarbonate complexes are not related to the direct formation of carbonyl and carboxyl groups on the catalyst since a higher temperature is needed for their formation. Considering their results, the group proposed that on the graphite-like carbon, firstly, the formation of epoxides and quinones takes place which further re-arrange to carbonates and lactones by interaction with O and $O_3$. Due to its high activity, atomic O is converted into carbonyls, while $O_3$ promotes the formation of lactones and carbonates. At higher temperatures, $CO_2$ is fully converted on the metal catalyst. The moieties produced on the metal catalyst are then interacting with the surface of the carbon. Various basic and acidic groups (along with $CO_2$) are generated on the carbon surface. The final products from carbon oxidation via plasma reaction are disrupted and functionalized carbon rings (see Figure 4). Similar mechanisms can be proposed for oxidation in atmospheric conditions (corona discharge) where ionized species of oxygen and nitrogen are formed.

In the work by Erden et al. [124], atmospheric plasma oxidation (i.e., corona treatment) of carbon fibres was performed, intending to enhance its surface chemistry to optimize the adhesion between carbon fibres and thermoplastics. The authors performed a so-called "sizing of the material" which makes the carbon fibre compatible with a coated layer of epoxy resin that protects the fibres from potential damage during transport and handling [11]. To improve adhesion between resin and carbon material, atmospheric

plasma oxidation was applied as primary functionalization. With a focus on changing carbon surface acidity, wettability and surface polarity, the authors [124] performed plasma oxidation at different exposure times. It is shown that already short residence times result in a significant surface oxidation, with increasing the oxygen-to-carbon ratio. Since the experiments were conducted under air, some nitrogen groups (amides) were incorporated at the carbon surface along with oxygen groups (alcohols/ethers, ketones, esters and carboxylic acids) as evidenced by XPS spectroscopy. Through incorporation of oxygen functionalities, both water wettability (fibre hydrophilicity) and surface tension were affected. These findings confirm that treating carbon materials under (oxidative) plasma conditions is a promising way to modify surface properties.

Pego et al. [110] modified the surface of activated carbon by corona treatment to explore its properties in adsorption. The authors focused on investigating corona treatment as a method for changing surface chemistry. This research group treated commercial activated carbon at various exposure times (up to 10 min) and observed its changes. It was proposed that exposure time may decrease electron density of the benzenoid structures from carbon by which reduction properties of the material are decreased. The authors explained that upon longer exposure, more oxygen groups are formed, e.g., carboxylic groups, and the carbon content of the surface decreases, which is probably caused by a damage in the structure. When discussing the type of oxygen groups, it is worth mentioning that these authors observed an increasing amount of oxygen groups for a long time exposure. Here, groups such as lactones and phenols depleted, and carboxylic groups were formed instead.

Summing up, both corona and plasma discharge are successful in the functionalization of various surfaces, including carbon surfaces [112,124–126]. Although the mechanisms of oxidation in plasma discharge are not fully understood, the process of plasma oxidation was applied in the functionalization of carbon nanotubes, as reviewed by Saka et al. [72]. In this paper, the performed plasma functionalizations are discussed, as well as various plasma reactors, plasma gases, different power of the systems and the influence of process parameters. This review is an excellent guidance for performing plasma oxidation of CNTs and other carbonaceous species.

### 3.2. Silane Coupling to Carbon Surfaces

Organosilanes form durable bonds with inorganic substrates and are employed as "bridge" (or compatibilizing agent) between an inorganic material and an organic matrix (such as a polymer of thermosetting resin). Functional organosilanes bear two characteristic functionalities:

$$X\text{-}R\text{-}Si\text{-}(OR')_n \quad n = 1, 2, 3 \ldots$$

Here, X stands for the organofunctional group, usually a non-hydrolyzable group (amino, vinyl, alkyl, epoxy etc.) that reacts with the polymer matrix. In some cases, X represents a non-polar group such as a perfluorinated alkyl chain to achieve, e.g., anti-wetting properties. R stands for a spacer and OR' stands for an easily hydrolysable group (alkoxy or acetoxy) that forms Si–OH groups which react with an inorganic substrate to form a covalent Si–O–Si bond [127]. Instead of -OR', a halogen, typically chlorine, can also be applied to achieve higher reactivity towards inorganic surfaces.

Organosilanes undergo two typical reactions: hydrolysis followed by condensation (Figure 5). Before reacting to each other or with a hydroxyl group on a substrate, up to three labile (OR') groups are hydrolysed and reactive silanol species are formed. Depending on the type of catalyst, hydrolysis can take place as an acid-catalysed or a base-catalysed reaction. Acid catalysed hydrolysis includes protonation of the leaving OR' group and the attack of water on the protonated organosilane via $S_N2$ mechanism (Figure 5a). Base-catalysed hydrolysis involves the attack of $OH^-$ on the organosilane to form an intermediate followed by bimolecular displacement of a leaving alkoxy or hydroxy group (Figure 5b). The following step is the formation of hydrogen bonds between oligomers and an (oxidized) substrate. Finally, covalent Si–O–Si links are formed and water is released as a byproduct (Figure 5c).

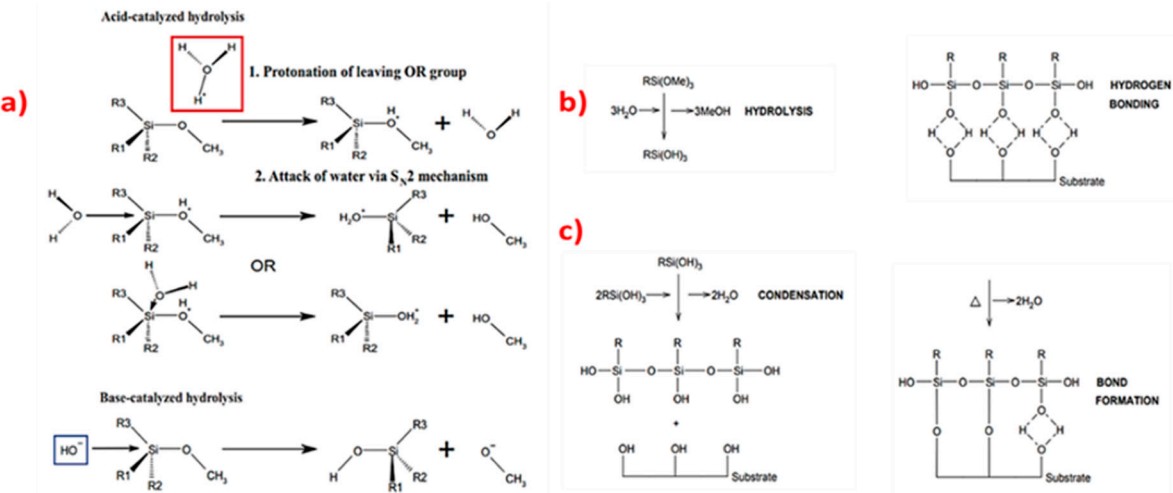

**Figure 5.** (**a–c**) Key reactions of organosilanes 2016, *Reproduced with permission from Dow Corning Corporation* [127] (*a colour version of this figure can be viewed online*).

Prior to this secondary surface functionalization, it is important to know how the material of interest (e.g., carbon) will react with silanes, what is the density of reactive sites on the carbon surface and which type of coverage is it aimed at (monolayer, multilayer or bulk). Monolayers are preferred when the organosilane structure on the treated surface must be uniform and when a uniform surface structure promotes specific interactions. The formation of monolayers is characteristic of monosubstituted organosilanes bearing only one hydrolysable substituent OR' on the Si atom. Moreover, mono- and bilayers are formed when the reaction of the organosilane with an OH-functional surface proceeds in a solvent such as toluene, which contains a maximum of 0.03 wt. % of water at 25 °C. An organosilane multilayer is less uniform and possesses improved hydrolytic stability between the adjacent phases. Because of the high density of the cross-linked silane network, a broken bond will not cause adhesive failure, as would be observed for monolayer coverage. Most providers of inorganic fillers (e.g., silica and alumina) have guides with all necessary information for the application of particular organosilanes, such as the surface area of filler (m$^2$/g) and its minimum coverage area (m$^2$/g). Minimum coverage area is related to the amount and type of present anchoring groups, e.g., hydroxyl, carbonyl, carboxyl, etc., on the primary functionalized substrates. The amount of required organosilane coupling agent (g) can be calculated from the following equation (Equation (1)) when used in systems with the inorganic substrate filler:

$$\text{Amount of silane (g)} = \frac{\text{Amount of inorganic filler (g)} \times \text{Surface area of filler }\left(\text{m}^2/\text{g}\right)}{\text{The minimum coverage area of the silane coupling agent }\left(\text{m}^2/\text{g}\right)} \tag{1}$$

From Table 1, where inorganic fillers are grouped according to their reactivity towards organosilanes, it can be concluded that the reaction between non-functionalized (i.e., neat) carbon materials and organosilanes is very poor. Thus, it is very important to find an adequate way to attach oxygen groups (preferably –OH) on the carbon surface prior to silanization. Generally, carbon materials (CNTs, carbon fibres, etc.) are used to reinforce polymers (thus obtaining composites).

**Table 1.** Reactivity between organosilanes and selected inorganic substrates [128].

| Adhesion Effect | Substrates |
|---|---|
| Excellent effect | Silica, alumina, glass, quartz, porcelain clay |
| Good effect | Mica, Talc, clay, water and alumina, grammiterion dust, potassium titanic acid |
| Slight effect | Asbestos, ferric oxides, zinc oxides, carborundum, silicon nitride |
| Poor effect | CaCO$_3$, BaSO$_4$, boron, **carbon** |

Even though carbon shows poor adhesion effects when mixed with organosilanes, a lot of results are found on the topic of carbon functionalization with glycydoxy and amino silanes, as these are highly compatible with commonly used epoxy and polyurethane resins. Kausar et al. [129] presented a review of amazing properties of silanized carbon materials, with reactive surface groups, e.g., oxygen groups, which were responsible for improved capabilities for mixing with polymer systems. Here, characteristics of epoxy-graphite, epoxy-graphene and epoxy-graphene nanoplatelet composites were discussed. The authors explained differences between these carbon materials and list the thermal and mechanical properties for composites prepared from each carbon material with epoxy resins.

In the following, some applications of organosilanes bearing (i) glycidyloxy, (ii) amino, (iii) mercapto and (iv) chloro substituents are highlighted. Besides the mostly used glycidyloxy organosilanes, aminosilanes have also found widespread applications for carbon surface silanization.

Silane coupling agents can also be used to make active carbon more prone to adsorption of various impurities from waste gases and wastewaters. Such an example is presented in a paper from He et al. [130], where the activated carbon surface was modified with 3-(glycidyloxypropyl)-trimethoxy silane (GPTMS) for an application in cigarette filters. Silane-functionalized carbon was used to remove smoke compounds. The authors performed oxidation and functionalization of activated carbon, and observed changes in surface area, surface porosity, volume and primary adsorption sites. A decreasing trend for these values was observed after oxidation, reduction and silanization of the carbon. To obtain more active (i.e., –OH) sites after oxidation, a reduction with hydrides was performed to convert carbonyl into hydroxyl groups. It was noticed that after reduction, the number of primary adsorption sites, e.g., –COOH decreased. In addition, these groups were converted into –OH. Both FT-IR and XPS results showed that coupling of silanes to the carbon surface had been successfully performed. Here, an increasing amount of silicon and oxygen were found at the surface. New C–O, Si–C and Si–O bonds were formed as evidenced by XPS spectroscopy. Regarding applications of the modified carbon, the authors carried out an experiment on water vapour adsorption in cigarette mainstream smoke. The authors concluded that with improved hydrophilicity, a trend of adsorption takes this order: reduced > oxidized > pristine > silanized carbon.

According to Aujara et al. [131], the functionalization of graphene oxide (GO) with organosilanes can be performed by gamma irradiation. In their work, 3-(aminopropyl)-triethoxysilane (APTES) and GPTMS were used as coupling agents between GO and a polymer matrix. To investigate the influence of gamma irradiation on modifying physicochemical properties of GO, GO in combination with silane solutions was exposed to $^{60}$Co gamma ray at different total doses. This type of radiation was used to explore a non-contact process in a mild reaction medium for the reduction of oxygen groups on prepared graphene oxide with silane coupling agents. After functionalization with amino silanes, characteristic FT-IR bands such as N–H, Si–C, Si–O–C and C–NH$_2$ were detected. The functionalization with glycidyloxy silanes gave comparable results, also proving the functionalization with organosilanes. The authors examined the morphology of their samples and noticed a roughened surface of functionalized GO which is subsequently less prone to agglomerate. This way of functionalization is a promising and novel method which is simpler to use, purer and less harmful than conventional reduction reactions.

Carbon is usually used as a filler in building materials [22–24,132–134] and the functionalization of carbon can additionally improve the mechanical properties of carbon-composites, which can be used as building material. Frequently, oxidation is the only step in the functionalization of a carbon surface prior to mixing it with cement-based materials [135–138]. Besides the formation of oxygen groups, various functionalities can be introduced such as fluorinated groups [138]. A non-covalent functionalization of carbon for mixing with cement-based materials was reviewed by Silvestro and Gleize [139]. Usually, it is related to dispersion methods such as combination of ultrasonication methods and use of additives based on polycarboxylate, naphthalene, sodium dodecyl sulphate and others.

An example of the functionalization of carbon with aminosilanes for applications in the building industry was presented by Silvestro et al. [140]. The authors functionalized multiwalled carbon nanotubes (MWCNTs) with APTES to overcome a problem of low interfacial interaction with cementitious matrix. They observed silanization effects on compressive strength after 1, 7 and 28 days. It was shown that the incorporation of 0.1% ($w/w$) of CNTs in cement increases its strength by 16% (for one-day-old cement composite) and by 13% (for 7- and 28-day-old cement composites). The research group concluded that this effect is a consequence of (i) reduced hydrophilicity of MWCNTs which can be seen in lower water adsorption capacity when mixing and enhanced hydration of cement, (ii) dispersing effect of silane agent and compactness of cement mixture and (iii) improved load transfer between the cementitious matrix and CNTs.

Wang et al. [141] reduced graphene oxide with mercaptosilanes for obtaining a functionalized surface for combination with a polymer matrix (in this case, poly-vinyl alcohol). The main goal of this research was to explore the properties of composites reinforced with graphene oxide (GO) and cellulose nanocrystals (CNCs) in the presence of highly amorphous poly-vinyl alcohol (HAVOH) that may find application in eco-friendly food packaging.

GO is significantly sensitive to moisture due to is hydrophilic properties, and the authors described different methods to reduce GO. They explored a double effect of silane coupling agents—as reductive agent and as cross-linking agent (via silane condensation).

The first effect is reducing GO groups by the nucleophilic opening of oxirane rings on the GO surface by the mercapto moiety of silane and the second effect is anchoring the GO to the HAVOH structure.

Regarding modified activated carbons with an application in adsorption, the paper from He et al. [142] presents an investigation on functionalizing an activated carbon surface (from coconut shells) with ionic liquids for $CO_2$ adsorption. Ionic liquids are used as adsorbents, but due to their high price and viscosity, their application on an industrial level is limited. To explore the advantages (low-cost and fast-diffusivity) of ionic liquid coatings on high-surface porous materials, the group performed an oxidation of activated carbon in nitric acid and subsequent functionalization with chlorosilanes. Chlorosilanes were grafted onto the carbon surface from a silane modified phosphonium ionic liquid. Both FT-IR and XPS results showed that both functionalization and coating of the surface were successfully performed. The new material consists of a porous surface and an ion exchanging layer. The ion exchanging layer is a phosphonium based ionic liquid by which gas adsorption selectivity ($CO_2/N_2$) of the carbon is improved. In addition, adsorbate penetration is highly affected by mass flow rate as well as intra-particle diffusion. Here, functionalities provided a higher amount of micropores, which strongly affects the adsorption process at the modified carbon structure. A lower blockage of the pores permits the absorption of $CO_2$ and blocks $N_2$ molecules.

Summing up, it has been shown that the functionalization of carbon with organosilanes can lead to improved properties and provides a well-established base for further applications in composite materials.

### 3.3. Other Chemical Functionalization Methods

As explained at the beginning of this chapter, the functionalization of a carbon surface is usually performed in a number of sequential steps: (i) pre-activation of the surface by promoting active sites, e.g., oxygen functionalities, (ii) functionalization with coupling agents and/or (iii) attaching monomers to the surface that undergo "grafting-from" reactions.

Although pre-oxidation of the surface is a fundamental step in tailoring carbon surface properties for further (secondary) functionalization, it suffers from several drawbacks such as rapidly roughened surfaces, altered crystal structure and a changed chemical composition. Most of the commonly used functionalization methods typically employ strong oxidizing agents such as $HNO_3$ which are responsible for damaging the carbon surface [143]. Consequently, different approaches for the functionalization of carbon have been sought. Many of these newly described methods are aiming at the functionalization

of carbon nanostructures such as graphene, CNTs and fullerenes, but are also applicable to other types of carbon such as pyrolytic carbon.

Generally speaking, graphene's unique thermal, electric and mechanical properties originate from its structure, including single-atom thickness and two-dimensional and extensive conjugation. These structural elements provide advantageous thermal, electric and mechanical properties. However, the application of graphene is challenging because of issues related to production, storage and processing as well as its chemical inertness.

As already mentioned, chemical modification can be achieved via either covalent or non-covalent interactions. Covalent functionalization of pristine graphene typically requires reactive species that can form covalent adducts with the $sp^2$ carbon structure in graphene. A covalent modification allows for fine control of its chemical structure and opening of the bandgap for electronic applications. The non-covalent modification methodologies are mostly based on, e.g., π–π stacking interactions and van der Waals forces, which are advantageous in preserving the pristine structure and properties [144].

In this section, only the strategies for the *chemical modification* of graphene and related types of carbon, the influence of modification and the applications in various areas are summarized. Graphene type and morphology, the nature of the substrate and mechanical forces are key factors affecting the reactivity of graphene. This is caused by differences in the basal plane and the under-coordinated edges of graphene, and the zig-zag versus arm-chair configurations [145]. Therefore, the stabilization and modification of graphene have attracted extensive interest.

According to Speranza [52] functionalization methods can be specified for various applications such as (i) increased solubility, (ii) biological applications, (iii) electrochemistry and (iv) energy storage as well as (v) composite materials. The reported methods comprise, e.g., ultrasound assisted exfoliation with polar/non-polar surfactants and organic solvents; the functionalization superficial -OH groups with, e.g., acetic anhydride to obtain acetyl groups; treatment with, e.g., aminobenzoic acids for immobilizing biomolecules; additional functionalization based on the π-π interactions; functionalization with various enzymes; and many more. The review of Speranza is a comprehensive manual for researchers interested in different approaches to the functionalization of carbon materials [52].

Regarding advanced methods [145] which have been described for the chemical surface modification of carbon, these can be grouped as follows:

- Addition of free radicals such as aryl radicals (Ar•) to benzenoid structures;
- Cycloaddition [1+2] of nitrene species to C=C bonds;
- Cycloaddition [1+2] of carbene species to C=C bonds;
- Cycloadditions [3+2] of, e.g., azomethine ylide to C=C bonds;
- Cycloaddition [4+2] of dienes to C=C bonds (Diels–Alder reaction);
- Reactions with superficial OH groups with of Meldrum's acid, acid chlorides and anhydrides;
- Producing a hydrogen terminated carbon surface (C–H) followed by generation of radicals (C•) and "grafting-from" reactions;
- Reaction with amines.

It should be mentioned that these methods are suitable for modifying carbons and nanocarbons on a laboratory scale but have limited applications for the large scale functionalization of carbon.

Free radicals have been employed to covalently modify graphene through addition, C-H insertion, or cycloaddition reactions as was reviewed by Park and Yan [108]. Free radicals additions can be generated, e.g., from diazonium salts and benzoyl peroxide [145]. Electron transfer from graphene to aryl diazonium ions and photofragmentation of benzoyl peroxide yield aryl radicals that are subsequently added to graphene and form covalent adducts. Another example of surface modification via radical coupling is described by Korivand and Zamani [146]. Here, electron deficient nitroaryl radicals (NO$_2$-Ar•) were grafted onto a carbon surface under microwave irradiation. The nitroaryl radicals (2-nitrophenyl, 4-nitrolhenyl, 2,4-dinitrophenyl and 2,4,6-trinitrophenyl) were generated in

situ by the reaction of nitroaniline derivatives with isoamyl nitrite as a diazozation agent, followed by the decomposition of nitroaryl diazonium intermediates. The authors found that nitroaryl radicals and graphene form a covalent bond. This provides the possibility to carry out reduction reactions yielding $NH_2$-Ar groups on the carbon surface which can be further converted by, e.g., the addition of isocyanates and isothiocyanates, and azo-coupling reactions. However, until now no such further modifications have been reported.

Another way of functionalization of carbon is based on nitrenes and carbenes which undergo [1+2] cycloaddition reactions to give aziridines and cyclopropanes, respectively. Nitrenes are electron-deficient species (generated by thermal or photochemical decomposition of aryl azides, Aryl-$N_3$) and they can efficiently functionalize graphene. The photochemistry of aryl azides to give singlet and triplet aryl nitrenes has been reviewed by Bou-Hamdan et al. [147]. Because perfluorophenyl nitrenes show less side reactions (e.g., ring expansion to azepines) compared to alkyl and phenyl nitrenes, perfluorophenyl azides are especially effective in surface modification [145]. The cycloaddition of nitrenes to C=C bonds has, e.g., been realized for the functionalization of exfoliated graphene sheets with azidophenylalanine (i.e., Boc–Phe(4-$N_3$)–OH), and yields hydroxyl terminated aziridine functionalities [104]. The hydroxyl group can be further modified, e.g., via the addition of isocyanates and other reagents. It should be noted that nitrenes also undergo C-H insertion as well as hydrogen abstraction reactions with hydrocarbons.

Carbenes have been less frequently described for carbon modification than nitrenes. Carbenes, e.g., C-$R_2$, undergo C-H insertion as well as [1+2] cycloaddition reactions with the C=C bonds where functionalities such as H, F, $CH_3$, CN, $NO_2$, $OCH_3$, CCH and $C_6H_5$ have been used to introduce named onto the graphene surface [148]. To generate carbenes, various reactions can be applied, which can be overviewed in an article by Frémont et al. [149]. A more recent method is based on the photolytic decomposition of diaziridines, see, e.g., references [150–152].

A typical example of [3+2] addition reactions to C=C bonds at the surface of carbon is related to the reaction of azomethine ylide. Azomethine ylide reacts in a 1,3-dipolar cycloaddition and has already been explored on various carbon nanostructures (e.g., COsC-NHs, fullerenes, CNTs) [102]. These authors [101] stated that this reaction offers flexibility in choosing desired precursors (aldehydes or amino acids) which offer the possibility of creating various functional groups.

One particular type of the covalent functionalization is an [4+2] addition of dienophiles to C=C bonds as presented in the review by Georgakilas et al. [153]. Except from free radicals, dienophiles can also react with $sp^2$ carbons on graphitic carbons. Examples for typical dienes are tetraphenylporphyrin, aldehydes, amino acids, phenyl azides and alkyl azides.

The generation of highly reactive intermediates in these reactions leads to side products that complicate the product composition and analysis. A particular feature of these [4+2] reactions is that the reaction is reversed at elevated temperatures, which has been exploited for self-healing of polymers [154]. The *retro* Diels–Alder reaction gives the educts again, which would cause a thermal instability of this type of surface modification. In addition, arynes can serve as a dienophile in a Diels–Alder type reaction with graphene (aryne cycloaddition) [155], see Figure 6.

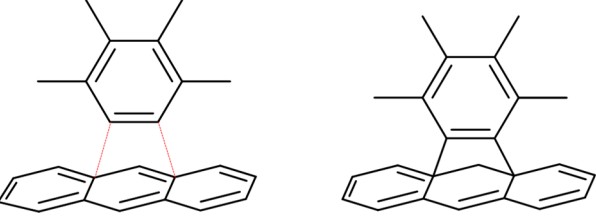

**Figure 6.** Functionalization of graphene with aryne group (i.e., cycloaddition of aryne group).

A bright example of (bulk) heterojunction with a carbon material has been presented by Sariciftci et al. [20,48,156,157] where a reversible photo-induced electron transfer was

observed for derivatives of buckminsterfullerene (β-carotene $C_{60}$) in the presence of intrinsically conductive polymers. Buckminster fullerene is an excellent electron acceptor and can form charge transfer salts with a variety of strong electron donors which makes $C_{60}$ a desired component in producing superconductive (polymer) materials. In this work, a composite was created from a conductive polymer poly [2-methoxy,5-(2′-ethyl-hexyloxy)-p-phenylene vinylene] (MEH-PPV) and fullerene. The created composite is very useful in photovoltaic cells based on bulk heterojunction. In these application, functionalized types of $C_{60}$ are applied, e.g., 1-(3-methoxycarbonyl) propyl-1-phenyl [6,6] $C_{61}$, also see references [158–160]. More recent methods on the functionalization of fullerenes with carbenes are provided in the review by Yamada et al. [161]. This application of a functionalized carbon type demonstrates that the chemical modification of carbon materials plays an important role in up-to-date technologies.

In the paper by Fedorczyk et al. [162], functionalized graphene was covalently bound to $C_{60}$ and $C_{70}$. Various Diels–Alder reactions can be performed on carbon surfaces. The functionalization of $C_{60}/C_{70}$ with graphene that has previously been modified with dianthracene malonate is one of them. This type of direct functionalization of graphene via a cyclopropanation with malonate (Bingel reaction) followed by the addition of $C_{60}$ produces the anthracene moieties (Figure 7). The result demonstrates that both, $C_{60}$ and $C_{70}$, fullerenes are capable of forming Diels–Alder adducts. This research led to the production of hybrid materials of graphene and fullerene with enhanced energy storage capability.

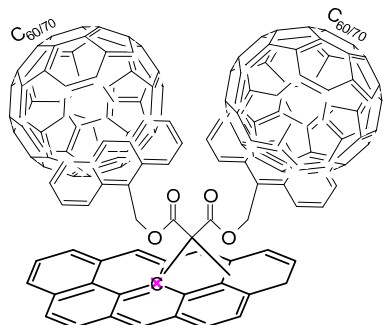

**Figure 7.** Functionalization of anthracene covered graphene with $C_{60}$ and $C_{70}$ via Diels–Adler reaction.

As reported in the paper by Ehlert et al. [143], grafting offers an alternative to oxidative functionalization, when a direct polymer grafting method is applied. The aim was to functionalize carbon fibres with carboxylic groups through grafting reactions by which the fibre tensile strength would be preserved, thus avoiding a damage of the surface by commonly used pre-oxidation methods. Here, isopropylidene malonate (Meldrum's acid) was used as an alternative for the grafting of –COOH groups. Meldrum's acid is an effective agent by which oxygen groups (phenols) can be easily converted to –COOH terminated ester groups (malonic acid ester) in ring opening reactions. It was concluded that the reaction of carbon with Meldrum's acid can graft ester moieties onto existing hydroxyl groups without preceding oxidation of the carbon, and that the morphology of carbon is preserved. In addition, it was proposed that this way of pre-functionalization is a good starting point for further grafting reactions.

Another method for the transformation of hydroxyl groups on the surface of a carbonaceous material is based on the acylation with either acid chlorides or anhydrides [163]. According to this reference, CNFs can be functionalized for the modification of cellulose food packing (i.e., cellulose nanofillers). Since CNFs have a high tendency to form agglomerates, the covalent bonding to cellulose is advantageous.

To overcome the problems of CNT aggregation and the poor dispersibility of CNTs in the polymer matrix (i.e., cellulose), both the carbon nanofiller and the cellulose material were acetylated using a mixture of acetic anhydride and acetic acid.

Another way of functionalization of graphene is based on the attachment of hydrogen ($H_2$). Graphane is a derivative that can be prepared from fully hydrogenated graphene

via low-temperature annealing. It was noted that well-structured hydrogenated graphene derivatives are interesting materials due to their magnetic, metallic and semiconducting properties [164].

The radical-induced grafting of terminal alkenes to carbon was discussed in the paper by Zhang et al. [164]. These researchers employed radical initiators that can selectively abstract hydrogen atoms from H-terminated carbon surfaces yielding carbon centred radicals (C•). As radical reactions benefit from less complex intermediates, the grafting of terminal alkenes using this approach was studied for four different carbon substrates (detonation nanodiamond, boron-doped diamond electrode, vitreous carbon planchet and citric acid-derived polymeric carbon dots). All of these hydrogenated carbon surfaces were, firstly, treated with benzoyl peroxide, a mild thermal initiator (dissociates between 80 °C and 95 °C) in order to avoid a rupture of the aromatic $sp^2$ structure. Second, alkenes such as 1H,1H,2H-perfluoro-1-octene, allyl trimethylammonium bromide and vinyl ethers were grafted onto the carbon surface. It was concluded that radical-initiated grafting of alkenes to H-terminated carbon surfaces provides extraordinary chemical stability and that this method avoids unwanted side reactions that are usually occurring when grafting is performed with light-induced reactions.

Another example of advanced functionalization of carbon surfaces is given by Moaseri et al. [165]. In their work, carbon fibres (CFs) were functionalized with aliphatic and aromatic diamines in order to improve interactions between the CFs and an epoxy resin. Here, unsized CFs were treated with aminobenzene (AB), ethylenediamine (EDA), 1,6-hexamethylenediamine (HAD) and 1,4-diaminobenzene (DAB) by using microwave irradiation in order to achieve different functionalization degrees. Microwave assisted functionalization was used to overcome low reaction rate and conversion efficiency. Unfortunately, no reaction mechanisms were presented for this type of surface functionalization. The authors reported that this process is not destructive for the structure of carbon and that it offers a high degree of surface functionalization.

All the above-mentioned examples of chemical functionalization of carbon materials are just a short insight in the extensive field of grafting methods. They present alternative approaches in designing a desired carbon surface which subsequently leads to the production of materials with modified properties. This is a confirmation that each functionalization method can be customized in such a way that for different carbon types additional properties can be obtained.

*3.4. Polymer-Based Carbon Composites*

Carbon composite materials have excellent properties and diverse applications. As explained in the previous chapters, surface modification is a key factor in preparing carbon materials for mixing with polymers. Organosilanes have been widely used for reinforcing composites as adhesion promoters, coupling agents, cross linkers and dispersing agents. Most of the published research is related to the functionalization of carbon fibres, but these procedures can also be applied to spheres and particles of carbon, also including pyrolytic carbon.

The compatibilization of carbon with polymers and resins is achieved by the proper selection of organofunctional groups, which can be reactive or non-reactive. Reactive groups such as epoxy, chloropropyl, vinyl, mercapto, disulfide, tetrasulfide, ureido or methacrylate are used to improve adhesion between systems. Non-reactive groups are mostly alkyl chains such as methyl, propyl, i-butyl, phenyl or n-octyl, and they are used as dispersing or hydrophobic agents [127]. Depending on which polymer (or resin) will be used, an appropriate organosilane coupling agent must be chosen. From Table 2, it can be seen which organosilanes are compatible with thermosetting resins.

Functionalized carbon materials confer improved mechanical and thermal properties to polymers, as it is known from carbon fibre composites. In addition, carbon can be used either as fillers dispersed in a polymer matrix, or the polymer itself can be used as a binder for carbonaceous materials. After successfully performing an oxidation of carbon, thus providing enough active sites (–OH) for silanes to attach, functionalized carbons are finally

ready to react with the polymer matrix. Carbons functionalized with organosilanes that are combined with epoxy or polyurethane resins are presented as promising materials to gain composites for diverse applications.

**Table 2.** Some of the most used thermosetting resins and suitable organosilane coupling agents [7–9].

| Thermosetting Resins | Silanes |
| --- | --- |
| **Epoxy** | Amine, epoxy, chloroalkyl, mercapto |
| Imide | Chloromethylaromatic, amine |
| Melamine | Amine, epoxy, alkanol amine |
| **Urethane** | Amine, mercapto, ureido, isocyanate |

3.4.1. Selected Matrix Resins for Carbon Composites

Numerous polymers can be chosen as matrix for carbon composites, comprising thermoplastic polymers, elastomers and thermosetting resins. Since the properties of the carbon–polymer composite do not only depend on the carbon material, but also on the properties of the matrix polymer (preferably based on epoxies and polyurethanes), these resins will be reviewed briefly in the following text.

Epoxy Resins

Epoxy resins (EP) are examples of thermosetting resins that have excellent features such as high strength level, long-term thermal and mechanical stability, low inflammability, smoke and toxicity, and excellent electrical and thermally insulating characteristics. In addition, epoxies possess excellent adhesion and good corrosion resistance and they are widely used in industrial applications (protective coatings, adhesives for various substrates, flooring, textiles, aerospace composites, marine and automotive structures, printed circuit boards, electrical insulation, etc.) [15,166]. EP are being widely used as polymer matrices and as binders for inorganic components. Depending on their structure, three types of EP can occur: cycloaliphatic, epoxidized and glycidylated compounds [12] (Figure 8). Cycloaliphatic epoxy resins provide higher UV stability and electrical properties because of their fully saturated structure compared to commonly used bisphenol epoxies (e.g., BADGE, also known as DGEBA) which are an example of glycidylated resins (Figure 8).

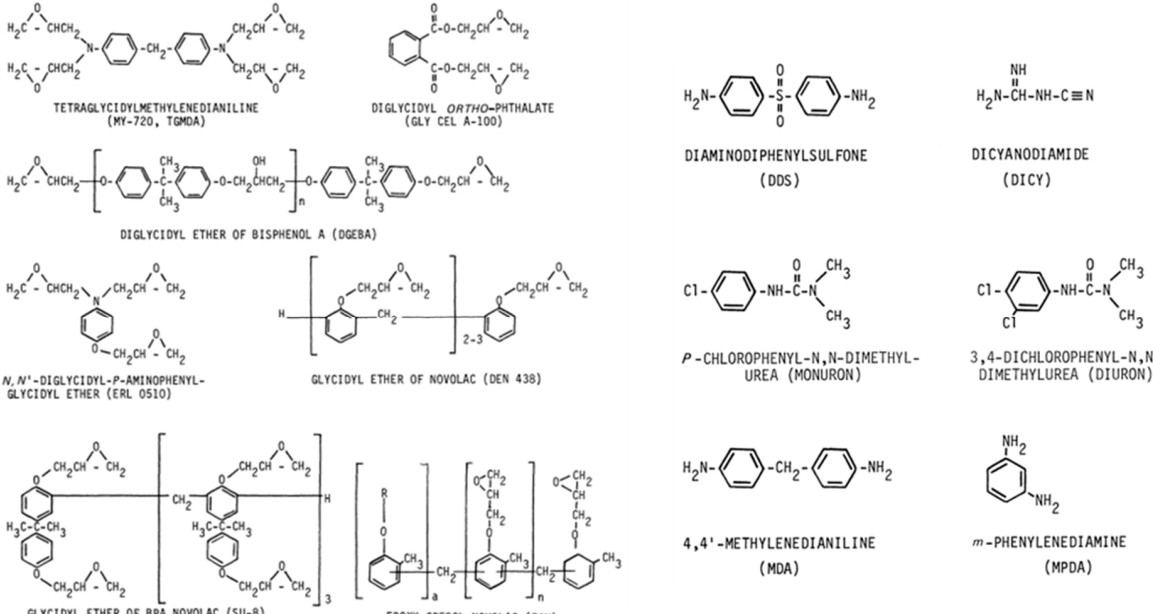

**Figure 8.** Common epoxy resins (**left**) and curing agents (**right**) 1987, Reproduced with permission from Taylor and Francis [13].

Commonly used processes for curing epoxies are room-temperature curing, thermal curing and photocuring. All of them require curing agents which are substances that (i) can add to the epoxy group or (ii) act as polymerization catalysts. Due to the diversity of curing agents, the obtained thermosets (and composites produced therefrom) differ in properties and chemical reactions. Curing agents for EP can be divided into three classes [13]: (i) active hydrogen compounds, which give a curing by a polyaddition reaction (polyamines, polyacids, polymercaptans, polyphenols etc.); ionic initiators that can be (ii) anionic (Lewis bases, e.g., tertiary amines, epichlorohydrin, ethylene oxide, propylene oxide) or (iii) cationic (Bronsted acids, Lewis acids such as halides of various metals, e.g., Sn, Al, Zn, B, Si, Fe, Ti, Mg, and Sb, as well as tetrafluoroborates of these metals). Furthermore (iv) cross-linkers can couple via the hydroxyl functionality of high molecular weight BPA epoxy resins. Figure 8 presents examples of common epoxy monomers and curing agents. Generally, the cured epoxy thermosets display high glass transition temperatures and brittleness as well. For further information on the chemistry and technology of epoxies, the reader is referred to, e.g., May [13].

Polyurethane Resins

Polyurethanes (PU) are one of the most investigated materials in the world next to polyolefins, PVCs, etc. They are characterized by high elasticity, abrasion resistance, low embrittlement, low-temperature resistance and good compatibility with metals and rubbers [14,166]. PU is produced from the reaction between polyols (polyether polyols and polyesters polyols, resp.) and di-/multifunctional isocyanates, although they can contain other functional groups as well (Figure 9). When considering polyols, these can be also divided by their molecular weight: high molecular weight polyols are used in flexible polyurethanes, while low molecular weight polyols are used in producing more rigid polyurethanes.

**Figure 9.** Common polyols (**left**) and isocyanates (**right**) for polyurethanes [14], 2016Reproduced with permission from Royal Society of Chemistry [13].

The most used isocyanates are methylene diphenyl diisocyanate (MDI), toluene di-isocyanate (TDI) and aliphatic diisocyanates such as hexamethylene diisocyanate (HMDI). Isocyanates start the curing of polyols via a polyaddition reaction in the presence of cata-lysts such as tertiary amines (e.g., 1,4-diazabicyclo[2.2.2]octane (DABCO)) and organotin components. In a review by Akindoyo et al. [14], the chemistry of polyurethanes, types, synthesis and applications are summarized. In their paper, the "click-chemistry" is high-lighted as an advanced way of synthesising these polymers. This refers to a group of reactions that have specific advantages such as a high reaction rate at relatively low tem-peratures, simplicity of use, ease of purification, versatility, high product yields and many more. Most of these demands are in line with Green Chemistry approaches by Anastas

and Warner [167]. As explained by Zhang et al. [168], click reactions proceed without the elimination of small molecules as byproducts, which is characteristic of polyaddition reactions. Click reactions are well suited for step-growth polymerizations of various polymers (linear, graft, hyper-branched, dendritic, etc.). Finally, polyurethanes can also be used in the surface functionalization of various carbon materials. Figure 9 displays selected polyols and diisocyanates, which are typically found in industrial systems.

### 3.4.2. Characteristic Examples of Carbon Composites with Epoxy Resins and Polyurethanes

In this section, some characteristic examples of carbon with matrix resins, in particular epoxies and PU resins, are described. However, no attempt is made to provide a comprehensive overview of carbon fibre reinforced resins and carbon fibre composites.

After solving the first obstacle in modifying the carbon surface, the poor miscibility with polymers needs to be overcome. Most frequently, functionalized carbons are added to polymer matrices due to their light weight and electrical and thermal conductivity.

As stated, the organofunctional group on silane dictates which type of resin can be used. Depending on the reactivity of the named groups, specific properties can be achieved. One of the ways to explain what is happening between the coupling agent and the polymer matrix is over a theory that explains diffusion between two present phases. After reacting with the substrate and between themselves, the attached organosilanes penetrate the present polymer. This phenomenon is termed inter-diffusion network (inter-penetrating network, IPN) and is characteristic of the interphase [127]. According to Plueddemann [169], the key factors that influence IPN are hydrolysis and condensation rate, the solubility parameters and the structural characteristics of the used materials. The IPN is a newly formed structure of two cross-linked polymers that are physically entangled but not chemically linked to each other [170]. That implies the formation of diffusive bonding through the polymer matrix, for example, hydrogen bonds between the functional group of the resins and the organofunctional group of the silane. In the following sections, recent research that explored novel ways in the functionalization of carbon and curing of carbon–polymer composites is reviewed.

The major drawback of traditional curing of thermosetting resins is the control of time and conversion. This led to the development of a new method, termed frontal photopolymerization (FP). As a starting point, photocuring is performed in surface layers of the resin in the presence of an adequate photo initiator; the released heat (polymerization enthalpy) then causes a thermal initiator to decompose and further achieves thermal curing. Most important, it is a self-sustaining reaction which provides very large depths of curing, also for carbon composite materials [16]. Different irradiations can be applied, such as infrared and UV light, as well as X-rays and electron beam radiation [17–19]. These are easier to control. Generally, FP lowers the time and energy compared to conventional thermal curing.

One of the main disadvantages of cured epoxies is high brittleness due to dense cross linking which also influences its strength and other mechanical and thermal properties (heat resistance and resistance to impact damage) [166]. By incorporating various fillers such as carbon materials [11,140,171–182] or organic silicon [4,165,182] both electrical and thermal conductivity, as well as adhesion and toughness of epoxy resins can be improved.

The paper by Lu et al. [183] investigated silanized carbon nanotubes as constituent in epoxy composites. Epoxy resins containing functionalized CNTs and photo-initiators were exposed to electron beam irradiation for curing. Increasing amounts of oxygen and silane functionalities at the CNT surface were noticed after oxidation and subsequent silanization, which is a proof that modification of the carbon surface was successfully performed. The authors showed that the hardness and tensile strength of these epoxy composites are better when compared to the unfilled resin. At higher concentrations of CNTs, the formation of agglomerates strongly affected the thermo-mechanical properties. This group presented electron beam radiation of silanized carbon/epoxy composites as an effective way of curing

since it requires a short time of only 44 min for curing (compared to traditional thermal curing of composites) and no usage of solvents.

Hoepfner et al. [174] investigated the influence of silane wrapping on CNTs. Carbon nanotubes are very well investigated as reinforcing material and, when functionalized, have a lower tendency to agglomerate. The focus of this paper was set on exploring the mechanical and thermal properties of epoxy nanocomposites with low concentrations of silane-functionalized CNTs. The authors observed that, after wrapping CNTs, there are no significant changes in the molecular structure. Covalent bonding of the silanes to the carbon surface was proven from FT-IR results where carbon–oxygen groups disappeared after silanization and new Si–O, Si–C and Si–O–C bonds were formed. The authors also showed that most of the CNTs are long and undamaged, which is proof that they are highly resistant to oxidation and that silanization affects re-agglomeration. When CNTs were added to epoxy resins at low concentrations, it was observed that the functionalized CNTs have lower thermal stability, which results from modification of the $sp^2$ structure to $sp^3$ by grafting functional groups. The resulting functionalized material displayed improved thermal properties when incorporated into an epoxy matrix. This research group accomplished the functionalization of carbon material with a low concentration of silanes, which improved the adhesion between carbon and the polymer matrix.

PU-carbon fibre composites were prepared and characterized by Borda et al. [184]. The authors investigated the properties of uncoated, epoxy and polyester coated CF in polyurethane resin. Changes were observed for Young's modulus and surface failure after fracture due to different arrangements of CF. To determine Young's modulus, the authors proposed a mathematical model by which the energy of influenced composite can be evaluated by considering the weight of added carbon and geometric properties in composite. In doing so, the changed equation for Young's modulus can predict linear trends with the rising amount of CF. The deviations in its behaviour are assumed to be linked with differences in the binding of CF, partial slipping within the matrix and the breaking of CF. Furthermore, coatings on the carbon surface exert an influence on the mechanical properties of the resulting composites: polyester resin coated CF seemed to have a polarity similar to that of the polyurethane matrix which results in higher adhesion of the matrix when compared to epoxy coated CF. With uncoated CF, the difference in polarity between the fibre and the polymer matrix was even more pronounced, but the highest fibre/matrix adhesion was obtained. This finding was attributed to the roughness of the uncoated CF surface where domains on the uncoated surface influence the transfer of stress onto the matrix.

In addition, another research group investigated the effect of CF surface silanization on CF reinforced polyurethane composites [173]. As already said, polyurethane resins are specific due to their distinguished elasticity, toughness, wear resistance and convenient change of hardness by choosing the desired ratio of hard to soft segments [185]. The end use of carbon–polymer composite is dictated by interfacial adhesion, and also depends on the homogenous distribution of CF in the polymer matrix. The research group used aminosilanes for silanization because of their good bonding with polyurethanes. XPS characterization of the CF showed characteristic signals of carbon, oxygen, nitrogen and silicon. Surface morphology showed that oxidized fibres have a more roughened surface compared to untreated CF, thus improving interfacial adhesion in the resulting PU/CF composites. After performing a step-by-step surface modification of the CFs (desizing, oxidation in nitric acid, reduction of carbonyl groups and subsequent silanization), the CFs showed increased polarity, wettability and roughness of the carbon surface, without a loss in tensile strength. The authors concluded that (i) homogenous dispersion and (ii) strong interfacial adhesion between the silanized CF surface and the PU matrix are key factors for improving mechanical properties.

The work of Fazeli et al. is aimed at higher strength and toughness of short CF/epoxy composites. Here the surface of CF was modified with a waterborne PU coating to increase interfacial bonding between the CF and the epoxy matrix resin [182]. This group used recycled

CF that can be technically obtained via pyrolysis, solvolysis and fluidized bed processing of CF composites. This group used silanization with APTES as a key step in the primary modification of the used CF. The authors proposed that immobilized APTES reacts with the isocyanate groups of the applied polyurethane coating, thus improving the flexibility and toughness of the coating. Summing up, carbon composites were produced with recycled CF. After surface functionalization of the recycled CF with APTES/polyurethane, the epoxy composites showed improved thermal stability, wettability and mechanical properties.

### 4. Conclusions

Finding the right application for carbonaceous residue (e.g., CB) from methane pyrolysis and pyrolysis of agricultural waste is one of the major topics in sustainability. Due to its propensity to form aggregates, low dispersity and contamination with heavy metals from catalysed reactions, the major challenge is finding a proper way to change and adapt the surface chemistry of the pyrolytic carbon. A proper fundament for functionalization is the generation of oxygen species at the surface of carbon. The chemical modification in various reagents at high temperatures has been used for many years as a flexible and easy way of oxidation. Due to environmental requests, new modifications such as hydrothermal and solvothermal oxidation as well as corona discharge and plasma oxidation are promising methods in changing the surface chemistry in a short period, with low energy consumption and reduced use of harmful chemicals. Besides traditional primary (e.g., oxidation) and secondary functionalization (e.g., silanization) approaches, additional chemical modification methods can be used, e.g., when there is high priority in preserving bulk materials' properties or performing reactions under specific conditions. In this article it is shown that there are specific and less-known chemical modification methods that can provide alternative routes for changing the carbon surface when traditional approaches are not applicable or not appropriate.

Although activated carbons are well-known for their diverse applications, the carbons from pyrolysis of gases or biomasses have a bright future in various fields such as the treatment of wastewaters and advanced applications in polymer composites, e.g., based on epoxy and polyurethane resins.

The functionalization of carbon surface is an alluring "playground" for all researchers who are seeking to upgrade and discover novel properties of (existing) materials. Especially applications that have a low impact on the environment and benefit from enhanced sustainability clearly deserve further research.

**Author Contributions:** Conceptualization, L.P. and W.K.; methodology, L.P.; software, L.P.; validation, L.P. and W.K.; formal analysis, L.P.; investigation, L.P.; resources, L.P.; data curation, L.P.; writing—original draft preparation, L.P.; writing—review and editing, W.K.; visualization, L.P.; supervision, W.K.; project administration, W.K.; funding acquisition, W.K. All authors have read and agreed to the published version of the manuscript.

**Funding:** This research received no external funding.

**Data Availability Statement:** The data presented in this study are available on request from the corresponding author.

**Acknowledgments:** L. Pustahija wishes to thank the Montanuniversität Leoben for funding her PhD thesis within a special research program.

**Conflicts of Interest:** The manuscript was written through the contributions of all of the authors. All authors have given approval to the final version of the manuscript. These authors contributed equally. The authors declare no conflict of interest.

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
