# Peer review of "Surface Functionalization of (Pyrolytic) Carbon—An Overview"

_carbon, 2023_

Round 1

Reviewer 1 Report

Journal: C

Ms. ID.: carbon-2197116

Title: Surface functionalization of (pyrolytic) carbon – an overview

Pustahija et al. aimed to present an overview of the properties of typical carbon materials with a focus on pyrolytic activated carbon. First, the most used primary and secondary functionalization methods are presented, together with the underlying reaction mechanisms. Finally, some characteristic applications of surface functionalized carbon are given.

It was a great pleasure to read this review. It is extremely useful. This topic is very important and suitable for the Journal. The manuscript is very well prepared. The introduction is short and informative. The balance between the sections is adequate. The conclusion is scientifically sound. All necessary information is contained. All important aspects are discussed. I have only one comment:

Why did the authors write “pyrolytic” carbon? I find quotation marks unnecessary.

Author Response

Dear Reviewer,
I would like to thank You for Your time and patience. Many thanks for Your kind and positive feedback on our article.

I would also like to answer Your question. Since throughout the whole article there weren´t specific methods of modifications for pyrolytic carbon-but for e.g., CNTs, activated carbon, graphene etc., we are assuming that all of these methods can be applied also to pyrolytic carbon. I hope that I managed to clarify our approach.

Once again,
Many thanks for Your time, patience and understanding.

With kind regards,
Lucija Pustahija

Reviewer 2 Report

The review describes various techniques for surface modification of carbon materials. The overall topic fits very well to the scope of this journal and a comprehensive review on this topic is interesting. However, the manuscript contains too many mistake and, very important for a review, a clear structure is not visible. Therefore, I have to recommend "rejection" of this manuscript. In the following, I would like to give a few comments that support my decision and hopefully help to improve this manuscript:

Generally, a clear structure is missing. It is not clear to which materials the term "pyrolytic carbon" refers. Is it just the carbonaceous residue from methane pyrolysis (as mentioned in the first sentence of the conclusion section) or also any kind of pyrolysed biomass? What are the adressed applications? The main chapter 3.1 seems more like general overview of carbon modification. A clear focus is missing. At the end there are even chapters about the preparation epoxy resins and polyurethanes.

Potential contamination with catalysts is mentioned as a main drawback. However, the presence of catalysts is not considered in any surface treatments.

The terms "activation" or "activated" seem to be used for oxidation or oxidized carbons, but also for highly porous carbons (e.g. activated carbons for wastewater treatment). This might lead to confusion.

In lines 97-98, energy source (coke), a component of alloys (steel) or a lubricant (graphite) are mentioned as major applications for carbons. -> Are any of those relevant for the following sections? What about carbon fibers for mechanical reinforcement, conducting fillers, various functional carbons in electrode materials, activated carbons for filters etc.? These seem to be much more relevant for the later chapters.

Line 99: Carbon exists in a few allotropes, such as graphite, diamond, and fullerene. -> There are many other allotropes. Are the mentioned ones really important for this review?

Long sections without any references: e.g. lines 50-61 or 92-104.

Line 128-129: ...high-value carbon black... [31-34] -> none of the cited references is describing carbon black. Reference 33 is about graphene & SWCNT, the other about activated carbon?!

In line 160, there is a link to chapter 0.

In chapter 3.1 is very unbalanced: e.g. in lines 208-214, 12 references are cited without any information about the content at all. In the following subchapters multiple paragraphs are used for one single reference. A clear focus is needed here. At the moment, it seems like a summary of carbon modification in general.

Line 577, Chapter 3.1.3: there are various studies using ozone treatment of carbon surfaces. These could be considered here.

Line 1146: ...next to polyolefins, polypropylene, PVCs, etc. -> polypropylene is a polyolefin

Lines 1464-1466: The authors are missing for references 68-69.

Author Response

Reviewer 2.

The review describes various techniques for surface modification of carbon materials. The overall topic fits very well to the scope of this journal and a comprehensive review on this topic is interesting. However, the manuscript contains too many mistake and, very important for a review, a clear structure is not visible. Therefore, I have to recommend "rejection" of this manuscript. In the following, I would like to give a few comments that support my decision and hopefully help to improve this manuscript:

Generally, a clear structure is missing. It is not clear to which materials the term "pyrolytic carbon" refers. Is it just the carbonaceous residue from methane pyrolysis (as mentioned in the first sentence of the conclusion section) or also any kind of pyrolysed biomass? What are the adressed applications? The main chapter 3.1 seems more like general overview of carbon modification. A clear focus is missing. At the end there are even chapters about the preparation epoxy resins and polyurethanes.

Answer to Reviewer 2:

Dear Reviewer,

I would like to thank You for the time to go through our article. Your comments were considered and the made modifications make our article better to understand. I would like to answer this comment as in general answer.
Firstly, we decided to keep the structure of the article, because our idea was to first deal with carbon in general and to put focus on pyrolytic carbon. The pyrolytic carbons here considered are the ones from the pyrolysis of CH4 and from biomass. These are examples on which we focused for an overview of functionalization methods-oxidation, which is mostly performed and silanization, and in the last step, incorporation of polymers into modified carbon in such a way.
Secondly, and linked to the previously mentioned, we are aware that the mentioned articles aren´t directly dealing with pyrolytic carbon, but we assume that these methods-both primary and secondary modifications can be applied to it (the short observation was added in several places throughout the article).
Thirdly, considering Your observation made in the last chapter of an overview of the epoxy and polyurethanes, we would like to mention that explanation of why is added was inserted at the beginning of the chapter.
All made changes are highlighted in yellow, and also, some Figures were changed. I appreciate Your suggestions and we hope that we fulfilled Your expectations. Lastly, the further answers for suggested changes are answered separately from this.

Once again,
Many thanks for Your time, patience and understanding.

With kind regards,
Lucija Pustahija

Potential contamination with catalysts is mentioned as a main drawback. However, the presence of catalysts is not considered in any surface treatments.

Answer: The catalytic oxidation wasn´t mentioned in the area in which we were focused. The short insert was made on this topic just by the Hummers and Offeman method where the presence of NaNO2 can be considered as a catalyst in this oxidation method.

The terms "activation" or "activated" seem to be used for oxidation or oxidized carbons, but also for highly porous carbons (e.g. activated carbons for wastewater treatment). This might lead to confusion.

Answer: The difference between activation and oxidation was clarified. We agree that this was needed.

In lines 97-98, energy source (coke), a component of alloys (steel) or a lubricant (graphite) are mentioned as major applications for carbons. -> Are any of those relevant for the following sections? What about carbon fibers for mechanical reinforcement, conducting fillers, various functional carbons in electrode materials, activated carbons for filters etc.? These seem to be much more relevant for the later chapters.

&

Line 99: Carbon exists in a few allotropes, such as graphite, diamond, and fullerene. -> There are many other allotropes. Are the mentioned ones really important for this review?

Answer: Due to the mentioned, this paragraph was overlooked and modified. Novel materials such as CNTs, CNRs, CO etc. were mentioned. The last version contained the Figure where the turbostratic structure was explained. Due to the modifications, this Figure was taken out. Many thanks for Your advice and observations, we found it really useful. 

Long sections without any references: e.g. lines 50-61 or 92-104.

Answer: This observation was also considered and references were added.

Line 128-129: ...high-value carbon black... [31-34] -> none of the cited references is describing carbon black. Reference 33 is about graphene & SWCNT, the other about activated carbon?!

Answer: I have taken Your observation into account and went through the references. This is my mistake and the right references are added.  

In line 160, there is a link to chapter 0.

Answer: due to the changes of our text to the template of the Journal, several mistakes of this kind occurred. This was updated and corrected.

In chapter 3.1 is very unbalanced: e.g. in lines 208-214, 12 references are cited without any information about the content at all. In the following subchapters multiple paragraphs are used for one single reference. A clear focus is needed here. At the moment, it seems like a summary of carbon modification in general.

Answer: Firstly, we would like to highlight that, although mentioned functionalization methods aren´t directly referring to pyrolytic carbon but other carbon materials-we assume that the latter can be applied to any carbon material-as well as on the pyrolytic carbon. Secondly, we considered Your suggestion of adding more references in this paragraph, and we hope that this looks better now.

Line 577, Chapter 3.1.3: there are various studies using ozone treatment of carbon surfaces. These could be considered here.

Answer: Due to the mentioned, the oxidation method with ozone was briefly elaborated and placed in the chapter along with the other dry oxidation methods. The text was accompanied by suitable references. We would like to highlight that we didn´t go too deep into this topic since we (i) focused on the dry oxidation methods like plasma (and corona discharge) (ii) we consider that ozone treatment (one of the AOPs is mostly used in the treatment of wastewaters, not carbon). I hope that You understand our approach and that inserting this short paragraph fulfilled Your expectations.

Line 1146: ...next to polyolefins, polypropylene, PVCs, etc. -> polypropylene is a polyolefin

Answer: Your observation was considered and the mistake was corrected.

Lines 1464-1466: The authors are missing for references 68-69.

Answer: Due to the problem in the referencing programme, this mistake occurred. This was considered and updated.

Reviewer 3 Report

The surface modification of the broad class of carbon materials provides the improvement of its performance, making it possible to introduce novel functionalities onto the surfaces. The manuscript under review focuses on the surface functionalization processes of pyrolytic graphite and polymer-based carbon composites, which usually have high concentrations of impurities and non-carbon inclusions. The presence of impurities does not greatly affect the main processing methods and the patterns of interaction between the carbon surface and the environment are preserved.

The review is written in sufficient detail, taking into account the chronology of the studies carried out, and is of interest to specialists working in this area of research. The review practically lacks a critical analysis of the results of studies conducted in the cited publications, but this does not detract from the value and informativeness of this manuscript.

It should be noted that this review is not unique in its kind. The surface modification of carbon materials has been repeatedly discussed in numerous reviews. Some of these reviews are mentioned in the bibliography. This manuscript would have benefited if the authors had supplemented the bibliography with some of these reviews. For example, A. Rehman, et al, Current progress on the surface chemical modification of carbonaceous materials // Coatings 2019, 9, 103; doi:10.3390/coatings9020103; R. Khan, et al, Covalent functionalization of carbon materials with redox-active organic molecules for energy storage // Nanoscale, 2021,13, 36-50 and others at the choice of authors.

Author Response

Dear Reviewer,
I would like to thank You for the time to go through our article. Many thanks for Your kind comments and positive feedback. We would like to say that we are aware that the article considers already mentioned topics, but here can be found (i) the proposed mechanisms of oxidation-both wet and dry at one place (ii) a general overview of the topic of the functionalization (with the highlight on oxidation and silanization-since they are commonly used) of different carbon materials and we assume that (iii) the mentioned method can be applied to the functionalization of the pyrolytic carbon.
We also considered adding the mentioned references in the article.

Once again,
Many thanks for Your time, patience and understanding.

With kind regards,
Lucija Pustahija

Round 2

Reviewer 2 Report

Dear authors

Thank you for considering the comments and providing a significantly improved version of the manuscript. I can suggest publication of this version.